# Hydrodynamics and multiscale order in confluent epithelia

**Josep-Maria Armengol-Collado[†], Livio Nicola Carenza[†], Luca Giomi***

Instituut-Lorentz, Leiden University, Leiden, Netherlands

**Abstract** We formulate a hydrodynamic theory of confluent epithelia: i.e. monolayers of epithelial cells adhering to each other without gaps. Taking advantage of recent progresses toward establishing a general hydrodynamic theory of *p*-atic liquid crystals, we demonstrate that collectively migrating epithelia feature both nematic (i.e. $p = 2$) and hexatic (i.e. $p = 6$) orders, with the former being dominant at large and the latter at small length scales. Such a remarkable multiscale liquid crystal order leaves a distinct signature in the system's structure factor, which exhibits two different power-law scaling regimes, reflecting both the hexagonal geometry of small cells clusters and the uniaxial structure of the global cellular flow. We support these analytical predictions with two different cell-resolved models of epithelia – i.e. the self-propelled Voronoi model and the multiphase field model – and highlight how momentum dissipation and noise influence the range of fluctuations at small length scales, thereby affecting the degree of cooperativity between cells. Our construction provides a theoretical framework to conceptualize the recent observation of multiscale order in layers of Madin–Darby canine kidney cells and pave the way for further theoretical developments.

## Editor's evaluation

This important work presents a hydrodynamic description of confluent epithelial monolayers that captures different forms of orientational order in a scale dependent fashion and couples this order with flows driven by active stresses. Solid evidence for the validity of this approach is provided by detailed numerical simulations of different model tissues. This work should be of interest to a broad range of biophysicists interested in tissue mechanics and active matter.

**\*For correspondence:**
giomi@lorentz.leidenuniv.nl

[†]These authors contributed equally to this work

**Competing interest:** The authors declare that no competing interests exist.

## Introduction

Collective cell migration – i.e. the ability of multicellular systems to cooperatively flow, even in the absence of a central control mechanism – has surged, in the past decade, as one of the central questions in cell biology and tissue biophysics (*Friedl and Gilmour, 2009*). Whether spreading on a synthetic substrate (*Serra-Picamal et al., 2012*) or invading the extracellular matrix (*Haeger et al., 2020*), multicellular systems can move coherently within their micro-environment and coordinate the dynamics of their actin cytoskeleton, while retaining cell–cell contacts. This ability lies at the heart of a myriad of processes that are instrumental for life, such as embryonic morphogenesis and wound healing, but also of life-threatening conditions, such as metastatic cancer.

Understanding the physical origin of this behavior inevitably demands reliable theoretical models, aimed at providing a conceptual framework for dissecting and deciphering the wealth of biophysical data stemming from in vitro experiments and in vivo observations. Following the pioneering works by *Honda, 1978*; *Nagai and Honda, 2001*; *Farhadifar et al., 2007*; *Bi et al., 2015*; *Bi et al., 2016*, and others (*Boromand et al., 2018*; *Mueller et al., 2019*; *Loewe et al., 2020*; *Monfared et al., 2021*), *cell-resolved* models have played so far the leading role in this endeavour. Taking inspiration from the physics of foams (*Graner et al., 2008*; *Marmottant et al., 2008*), these models portray a confluent tissue as a collection of adjacent or overlapping polygonal cells (*Figure 1a, b*), whose

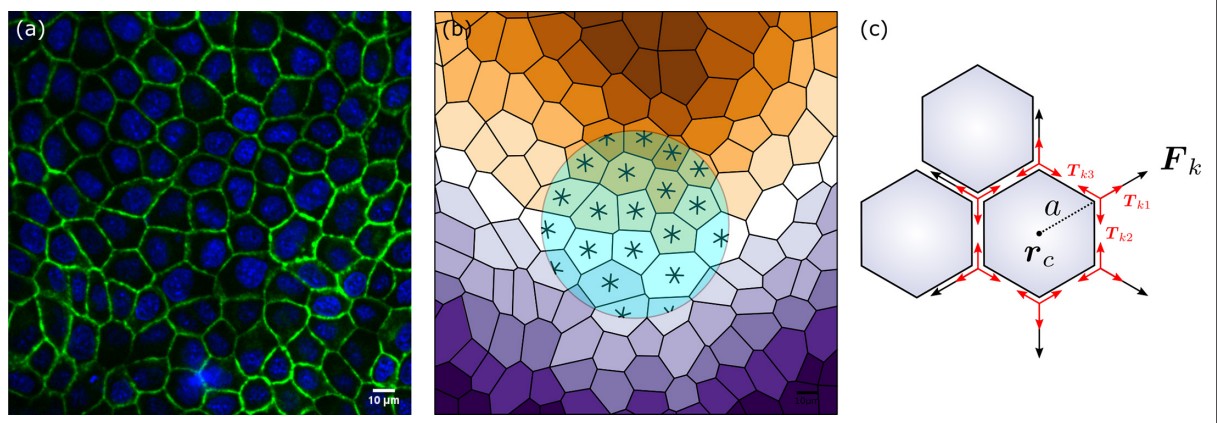

**Figure 1.** Epithelia and hexanematic order. (**a**) Example of multiscale hexanematic order in an in vitro layer of Madin–Darby canine kidney (MDCK) cells (**b**) and its computer-constructed segmentation. Both panels are adapted from Figure 3 of *Armengol-Collado et al., 2023*. The six-legged stars in the shaded region denote the sixfold orientation of the cells obtained using the approach summarized in Methods. The colored stripes mark the configuration of the nematic director at the length scale of the light-blue disk. (**c**) Schematic representation of the sixfold symmetric force complexion exerted by cells. The red arrows indicate the structure of the contractile forces acting within the cellular junctions.

dynamics is assumed to be governed by a set of overdamped Langevin equations, expressing the interplay between cells' autonomous motion and remodeling events, which change the local topology of the cellular networks.

Despite their conceptual simplicity, cell-resolved models agree remarkably well with experimental data on confluent monolayers (*Park et al., 2015*; *Atia et al., 2018*). In particular they account for a solid-to-liquid transition controlled by the cells velocity and their compliance to deformations (*Bi et al., 2015*; *Bi et al., 2016*; *Loewe et al., 2020*). Furthermore, as demonstrated by Pica Ciamarra and coworkers, the solid and isotropic liquid states of these model-epithelia are separated by an intermediate *hexatic* phase, in which the system exhibits the typical sixfold rotational symmetry of two-dimensional crystals and yet is able to flow (*Li and Ciamarra, 2018*; *Pasupalak et al., 2020*). Shortly after discovery, the same property has been recovered within the framework of the cellular Potts model, thereby strengthening the idea that hexatic order may in fact serve as a guiding principle to unravel the collective dynamics of confluent epithelia (*Durand and Heu, 2019*). Furthermore, recent in vitro studies of Madin–Darby canine kidney (MDCK) cell layers demonstrated that epithelial layers can in fact feature both nematic and hexatic orders, with the former being dominant at large and the latter at short length scales (see *Figure 1a, b* and *Armengol-Collado et al., 2023*; *Eckert et al., 2023*). This remarkable example of physical organization in biological matter, referred to as multiscale *hexanematic* order in *Armengol-Collado et al., 2023*, is believed to complement the complex network or regulatory pathways available to individual cells to achieve multicellular organization and select specific scale-dependent collective migration strategies.

Motivated by these recent discoveries, in this article we propose a continuum theory of confluent epithelia rooted in the hydrodynamics of liquid crystals with generic *p*-atic rotational symmetry (hereafter *p*-atic liquid crystals). Previous theories of epithelial hydrodynamics can be schematically grouped in two categories: (1) models based on (isotropic/polar/nematic) active gels (*Ranft et al., 2010*; *Popović et al., 2017*; *Pérez-González et al., 2019*); (2) models built around the so-called shape tensor (*Ishihara et al., 2017*; *Czajkowski et al., 2018*; *Hernandez and Marchetti, 2021*; *Grossman and Joanny, 2022*), i.e. a rank-2 tensor, similar to the inertia tensor in kinematics, that embodies the geometrical structure of the polygonal cells. Although both classes of models hold great heuristic value and represent a solid foundation for any future development, they suffer from the same limitation: being based on a tensorial order parameter whose rank is two or less, they can account at most for twofold rotational symmetry (i.e. nematic order), while leaving the small-scale hexatic order unresolved. To overcome this limitation, here we exploit recent advances toward extending the classic hydrodynamic theory of hexatic liquid crystals (*Zippelius, 1980*; *Zippelius et al., 1980*) to account for arbitrary *p*-fold rotational symmetry order (*Giomi et al., 2022a*; *Giomi et al., 2022b*), with $p = 2$ and $p = 6$ being the most relevant cases (but possibly not the only) in the context of epithelial dynamics.

We demonstrate that multiscale order is inherent to *active* liquid crystals with coupled order parameters, because of the indissoluble connection between shape and forces characterizing this class of non-equilibrium systems. Using fluctuating hydrodynamics, we explicitly compute the structure factor of epithelial layers and unveil a fascinating interplay between the nature of momentum dissipation (i.e. viscosity or friction) and noise at short length scales, where hexatic order is dominant. Such a mechanism profoundly affects the range of density fluctuations and could be harnessed to control the degree of collectiveness of cellular motion. Finally, by testing predictions against two different microscopic models of epithelia we demonstrate the robustness of multiscale hexanematic order across the rich landscape of models of epithelia.

## Results and discussion

### The model

Two-dimensional *p*-atic liquid crystals are traditionally described in terms of the orientation field $\psi_p = e^{ip\vartheta}$, with $\vartheta$ the local orientation of the *p*-fold mesogens. A more general approach, proposed in *Giomi et al., 2022a*; *Giomi et al., 2022b* and especially suited for hydrodynamics, revolves instead around the rank-*p* tensor order parameter, $\boldsymbol{Q}_p = Q_{i_1 i_2 \cdots i_p} \boldsymbol{e}_{i_1} \otimes \boldsymbol{e}_{i_2} \otimes \cdots \otimes \boldsymbol{e}_{i_p}$ with $i_n = \{x, y\}$ and $n = 1, 2 \ldots p$, constructed upon averaging the *p*th tensorial power of the local orientation $\boldsymbol{\nu} = \cos\vartheta \, \boldsymbol{e}_x + \sin\vartheta \, \boldsymbol{e}_y$. That is

$$\boldsymbol{Q}_p = \sqrt{2^{p-2}} \left[\!\left[ \langle \boldsymbol{\nu}^{\otimes p} \rangle \right]\!\right] = \sqrt{2^{p-2}} |\Psi_p| \left[\!\left[ \boldsymbol{n}^{\otimes p} \right]\!\right] , \tag{1}$$

where $\langle \cdots \rangle$ denotes the ensemble average and the operator $\|...\|$ has the effect of rendering an arbitrary tensor traceless and symmetric (*Hess, 2015*). The vector $\boldsymbol{n} = \cos\theta \, \boldsymbol{e}_x + \sin\theta \, \boldsymbol{e}_y$ is the analog of the director field in standard lexicon of nematic liquid crystals and marks the average cellular direction, which in turn is invariant under rotations of $2\pi/p$. The fields $|\Psi_p|$ and $\theta$ represent, respectively, the magnitude and phase of the complex *p*-atic order parameter $\Psi_p = \langle \psi_p \rangle$, while the normalization factor is chosen so that $|\boldsymbol{Q}_p|^2 = |\Psi_p|^2/2$ for all *p* values. For $p = 2$, *Equation (1)* readily gives the standard nematic order parameter tensor: i.e. $\boldsymbol{Q}_2 = |\Psi_2|(\boldsymbol{n} \otimes \boldsymbol{n} - \mathbb{1}/2)$, with 1 the identity tensor. In practice, if a cell's planar projection consists of a regular *p*-sided polygon, the microscopic orientation $\vartheta$ equates that of any of the vertices of the polygon. In the more realistic case of an *irregular* polygon, on the other hand, $\vartheta$ is given by the phase of the complex function $\gamma_p$, arising form the *p*-fold generalization of the classic *shape tensor* (*Aubouy et al., 2003*). This function was introduced in *Armengol-Collado et al., 2023* and is reviewed in Methods for sake of completeness.

The order parameter tensor $\boldsymbol{Q}_p$, the mass density $\rho$, and the momentum density $\rho\boldsymbol{v}$, with $\boldsymbol{v}$ the local velocity field, comprise the set of hydrodynamic variables describing the dynamics of a generic *p*-atic fluid, which in turn is governed by the following set of partial differential equations (*Giomi et al., 2022a*; *Giomi et al., 2022b*):

$$\frac{D\rho}{Dt} + \rho\nabla \cdot \boldsymbol{v} = (k_{\mathrm{d}} - k_{\mathrm{a}})\rho , \tag{2a}$$

$$\rho\frac{D\boldsymbol{v}}{Dt} = \nabla \cdot \boldsymbol{\sigma} + \boldsymbol{f}, \tag{2b}$$

$$\frac{D\boldsymbol{Q}_p}{Dt} = \Gamma_p \boldsymbol{H}_p + p \left[\!\left[ \boldsymbol{Q}_p \cdot \boldsymbol{\omega} \right]\!\right] + \bar{\lambda}_p \mathrm{tr}(\boldsymbol{u})\boldsymbol{Q}_p + \lambda_p \left[\!\left[ \nabla^{\otimes(p-2)}\mathbf{u} \right]\!\right] + \nu_p \left[\!\left[ \nabla^{\otimes(p \,\mathrm{mod}\, 2)}\mathbf{u}^{\otimes \lfloor p/2 \rfloor} \right]\!\right] \tag{2c}$$

where $D/Dt = \partial_t + \boldsymbol{v} \cdot \nabla$. *Equation (2a)* and *Equation 2b* are the mass and momentum conservation equations, with $k_{\mathrm{d}}$ and $k_{\mathrm{a}}$ rates of cell division and apoptosis, $\boldsymbol{\sigma}$ the stress tensor and $\boldsymbol{f}$ an arbitrary external force per unit area. In *Equation (2c)*, $\Gamma_p^{-1}$ is a rotational viscosity and $\boldsymbol{H}_p = -\delta F/\delta \boldsymbol{Q}_p$ is the molecular tensor describing the relaxation of the *p*-atic phase toward the minimum of the free energy $F$ (see Methods). The rank-2 tensors $\boldsymbol{\omega} = [\nabla\boldsymbol{v} - (\nabla\boldsymbol{v})^{\mathsf{T}}]/2$ and $\boldsymbol{u} = [\nabla\boldsymbol{v} + (\nabla\boldsymbol{v})^{\mathsf{T}}]/2$, with $\mathsf{T}$ indicating transposition, are the vorticity and strain rate tensors, respectively, whereas the dot product in the first line of the equation implies a contraction of one index of $\boldsymbol{Q}_p$ with one of $\omega$: i.e. $(\boldsymbol{Q}_p \cdot \boldsymbol{\omega})_{i_1 i_2 \cdots i_p} = Q_{i i_2 \cdots i_p} \omega_{j i_p}$. On the second line $(\nabla^{\otimes n})_{i_1 i_2 \cdots i_n} = \partial_{i_1} \partial_{i_2} \cdots \partial_{i_n}$, while $\lfloor \ldots \rfloor$ denotes the floor function and $p \mod 2 = p - 2\lfloor p/2 \rfloor$ is zero for even *p* values and one for odd *p* values. Finally, $\bar{\lambda}_p$, $\lambda_p$, and $\nu_p$ are material parameters expressing the strength of the coupling between *p*-atic order and flow.

Now, in order for *Equation 2a* to account for the dynamics of epithelial cell layers, we must specify the structure of the external force $f$ in *Equation 2b* and the stress tensor $\sigma$. As cells collectively crawl on a substrate, at a speed of order 0.1–1 µm/min (*Brugués et al., 2014*; *Angelini et al., 2011*), the former can be model as a Stokesian drag: $f = -\varsigma v$, with $\varsigma$ a drag coefficient. A more realistic treatment of the interplay between the cells and the substrate would account for the traction forces exerted by the cells' cryptic lamellipodium as well as for the compliance of the substrate (*Trepat et al., 2009*) and will be considered in the future. The stress tensor, on the other hand, is routinely decomposed into a passive and an active component: i.e. $\sigma = \sigma^{(\mathrm{p})} + \sigma^{(\mathrm{a})}$. The passive stress tensor is in turn expressed as $\sigma^{(\mathrm{p})} = -P\mathbb{1} + \sigma^{(\mathrm{e})} + \sigma^{(\mathrm{r})} + \sigma^{(\mathrm{v})}$, where $P$ is the pressure, $\sigma^{(\mathrm{e})}$ is the *elastic* stress, arising in response of a static deformation of a fluid patch, and $\sigma^{(\mathrm{r})}$ and $\sigma^{(\mathrm{v})}$ are, respectively, the *reactive* (i.e. energy preserving) and *viscous* (i.e. energy dissipating) stresses originating from the reversible and irreversible couplings between *p*-atic order and flow. The generic expression of $\sigma^{(\mathrm{p})}$ was derived in *Giomi et al., 2022b* and is reported in Methods.

The active stress $\sigma^{(\mathrm{a})}$, on the other hand, can be constructed phenomenologically for arbitrary $p$ values in the form

$$\sigma^{(\mathrm{a})} = \sum_{p} \left( \alpha_p \nabla^{\otimes (p-2)} \odot Q_p + \beta_p \left[\!\left[ \nabla^{\otimes 2} |Q_p|^2 \right]\!\right] \right), \tag{3}$$

where the symbol $\odot$ denotes a contraction of all matching indices of the two operands and yields a tensor whose rank equates the number of unmatched indices: i.e. letting $A_p$ and $B_q$ be two generic tensors of rank $p < q$, then $(A_p \odot B_q)_{i_1 i_2 \ldots i_{q-p}} = A_{j_1 j_2 \ldots j_p} B_{j_1 j_2 \ldots j_p i_1 i_2 \ldots i_{q-p}}$. The sum over $p$, finally, reflects the possibility of having not only one, but multiple types of *p*-atic order coexisting within the same system, as experiments on in vitro layers of MDCK cells have recently suggested (*Armengol-Collado et al., 2023*; *Eckert et al., 2023*).

Before exploring the consequences of the latter assumption, some comment about the physical interpretation of the terms featured in *Equation 3* is in order. The first term on the right-hand side of *Equation 3* is the stress resulting from the contractile or extensile forces exerted at the length scale of individual cells. To illustrate this concept one can assume each cell to exert a *p*-fold symmetric force complexion: i.e. $F_c = \sum_{k=1}^{p} F_k \delta(r - r_c - a\nu_k)$ with $F_k$ the force exerted by a cell at each vertex and originating from the imbalance of the tensions $T_{kl}$, driven by the active contraction of the cellular junctions, converging at the $k$th vertex: i.e. $F_k = \sum_l T_{kl}$ (see *Figure 1c*). The quantities $r_c$ and $a$ are the cell's centroid and circumradius, respectively, while $\nu_k = \cos(\vartheta + 2\pi k/p)\, e_x + \sin(\vartheta + 2\pi k/p)\, e_y$. We stress that, while the individual tensions acting along the junctions are exclusively contractile, the resulting vertex forces can be either contractile (i.e. $F_k \cdot \nu_k < 0$) or extensile ($F_k \cdot \nu_k > 0$), depending on the overall tension distribution and the geometry of the cellular network. Next, assuming $F_k = f\nu_k$ and expanding the delta function about $a = 0$ yields $F_c = \sum_{m=0}^{\infty} f_m$, where

$$f_m = \nabla^{\otimes m} \odot \left[ \frac{(-a)^m f}{m!} \left( \sum_{k=1}^{p} \nu_k^{\otimes (m+1)} \right) \delta(r - r_c) \right]. \tag{4}$$

Because of the *p*-fold symmetry of the force complexion $f_m = 0$ for all even $m$ values, unless $m = p - 1$, whereas odd $m$ values yields, up to symmetrization, $\sum_{k=1}^{p} \nu_k^{\otimes (m+1)} \sim \mathbb{1}^{\otimes (m+1)/2}$. Thus, after some algebraic manipulation, one finds $F_c \approx -apf/2 \nabla[(1 + a^2/8\, \nabla^2 + \cdots)\delta(r - r_c)] + f_{p-1}$. Finally, taking $\langle \sum_c F_c \rangle = -P^{(\mathrm{a})}\mathbb{1} + \sigma^{(\mathrm{a})}$ gives the following expression for contributions to the pressure and the deviatoric stress resulting from the active expansion and contraction of the cells. That is

$$P^{(\mathrm{a})} = \frac{apf}{2} \left( n + \frac{a^2}{8} \nabla^2 n + \cdots \right), \tag{5a}$$

$$\sigma^{(\mathrm{a})} = \frac{(-a)^{p-1} pnf}{\sqrt{2^{p-2}}\,(p-1)!} \nabla^{\otimes (p-2)} \odot Q_p, \tag{5b}$$

where $n = \langle \sum_c \delta(r - r_c) \rangle$ is the cell number density. From *Equation 5b*, one finds the following expression for the phenomenological parameter $\alpha_p$ in *Equation 3*: i.e. $\alpha_p = (-a)^{p-1} pnf/[\sqrt{2^{p-2}}\,(p-1)!]$. Notice that both constants $a$ and $f$ involved in *Equation 5a* are, in general, order dependent. We will come back on this aspect in Conclusion.

The second term in *Equation 3*, in contrast, expresses the active stress resulting from the spatial variations of the *p*-atic order parameter and, although similar to other contributions to the passive stress $\boldsymbol{\sigma}^{(\mathrm{p})}$, cannot be derived from equilibrium considerations. Other terms constructed by contracting $\boldsymbol{Q}_p$ with $\nabla^{\otimes 2}$ can be expressed as linear combinations of this and $\boldsymbol{\sigma}^{(\mathrm{p})}$, thus lead to a mere renormalization of the material parameters. It must be noted that the stress tensor enters in *Equation 2b* only via its divergence. Thus, possible second-order active terms such as $Q_{k_1 k_2 \cdots k_p} \partial_i \partial_j Q_{k_1 k_2 \cdots k_p}$, $Q_{ijk_3 \cdots k_p} \partial_{l_1} \partial_{l_2} Q_{l_1 l_2 k_3 \cdots k_p}$, etc., are mechanically equivalent to the terms $\partial_i Q_{k_1 k_2 \cdots k_p} \partial_j Q_{k_1 k_2 \cdots k_p}$ and $Q_{k_1 k_2 \cdots i} H_{k_1 k_2 \cdots j} - H_{k_1 k_2 \cdots i} Q_{k_1 k_2 \cdots j}$ arising from the passive stresses, as both sets of terms lead to the same body forces.

We observe that *Equation 3* already entails a multiscale hydrodynamic behavior even when a single $p$ value is considered. Such a crossover is expected at length scales larger than $\ell = (\alpha_p / \beta_p)^{1/(p-4)}$, where the second term of the right-hand side of *Equation 3* overweights the first term, reflecting the $p$-fold symmetry of the local active forces. In the presence of multiple types of $p$-atic order, the $p$-dependent structure of the active stress renders the multiscale nature of the system enormously more dramatic. To illustrate this crucial point, here we postulate the system to behave as a hexanematic liquid crystal. Formally, such a scenario can be accounted by simultaneously solving two variants of *Equation 2c*, for $\boldsymbol{Q}_2$ and $\boldsymbol{Q}_6$. In turn, the interplay between nematic and hexatic order results from a combination of dynamical and energetic effects. The former arise from active flow, which affects the local configuration of both tensor order parameters via the last four terms in *Equation 2c*. The latter, instead, can be embedded into the free energy $F = \int \mathrm{d}A\,(f_2 + f_6 + f_{2,6})$, where

$$f_p = \frac{1}{2} L_p |\nabla \boldsymbol{Q}_p|^2 + \frac{1}{2} A_p |\boldsymbol{Q}_p|^2 + \frac{1}{4} B_p |\boldsymbol{Q}_p|^4 \,, \tag{6a}$$

$$f_{2,6} = \kappa_{2,6} |\boldsymbol{Q}_2|^2 |\boldsymbol{Q}_6|^2 + \chi_{2,6}\, \boldsymbol{Q}_2^{\otimes 3} \odot \boldsymbol{Q}_6 \,. \tag{6b}$$

Here, $A_p$ and $B_p$ are constants setting the magnitude of the order parameter at the length scale of the short distance cut-off, here assumed to be of the order of the cell size, and $\kappa_{2,6}$ determines the extent to which the magnitude of the hexatic order parameter is influenced by that of the nematic order parameter and vice versa. The constant $\chi_{2,6}$, on the other hand, is analogous to an inherent susceptibility, expressing the propensity of the nematic and hexatic directors toward mutual alignment. The free energy contribution $f_{2,6}$ can further be augmented with several additional terms of higher differential order: e.g. $(\boldsymbol{Q}_2 \odot \nabla \boldsymbol{Q}_2) \cdot (\boldsymbol{Q}_6 \odot \nabla \boldsymbol{Q}_6)$, $|\nabla (\boldsymbol{Q}_2^{\otimes 3} \odot \boldsymbol{Q}_6)|^2$, $\nabla^2 (\boldsymbol{Q}_2^{\otimes 3} \odot \boldsymbol{Q}_6)$, etc. For simplicity, here we ignore these and higher-order couplings and focus on the zeroth order terms included in *Equation 6b*.

Crucially, *Equation 3*, *Equation 6a* entail two length scales, reflecting the distance at which the passive torques originating from the entropic elasticity of the nematic and hexatic phases counterbalance those arising from the active stresses:

$$\ell_2 = \sqrt{\frac{L_2}{|\alpha_2|}} \,, \qquad \ell_6 = \sqrt{\frac{|\alpha_6|}{L_6}} \,. \tag{7}$$

The former is the well-known active nematic length scale, dictating both the hydrodynamic stability (*Voituriez et al., 2005*) and the large-scale structure of spatiotemporal chaos in active nematics (*Giomi, 2015*) and whose signature in multicellular systems has been identified in both eukaryotes (*Blanch-Mercader et al., 2018*) and prokaryotes (*You et al., 2018*). The latter, on the other hand, sets the typical size of hexatic domains at the small length scale. Remarkably, $\ell_2$ and $\ell_6$ inversely depend on the magnitude of cellular forces (see *Equation 5a*). Thus, increasing activity has the effect of collapsing the multiscale structure of the system toward a single length scale, where $\ell_2 \approx \ell_6$. Two additional length scales, of purely passive nature, originate from the competition between rotational diffusion and the ordering dynamics driven by either liquid crystalline structure on the other one. These are given by $\ell_{\chi,2} = \sqrt{L_2/\chi_{2,6}}$ and $\ell_{\chi,6} = \sqrt{L_6/\chi_{2,6}}$. Their role will be discussed in the following section, in the framework of fluctuating hydrodynamics.

Finally, in the passive limit, when $\alpha_2 = 0$ and $\alpha_6 = 0$, *Equations 2 and 6*, reduce to those of a two-dimensional liquid crystal with coupled nematic and hexatic order parameter. The latter can be found, e.g., in free-standing liquid hexatic films (*Dierker and Pindak, 1987*; *Sprunt and Litster, 1987*), where molecules are either orthogonal to the mid-surface of the film or tilted by a fixed angle. In the latter case, the projection of the average molecular direction on the tangent plane of the mid-surface gives rise to in-plane nematic order, which is coupled to the sixfold *bond*-orientational order

associated with the underlying hexatic phase (see e.g. *Bruinsma and Aeppli, 1982*; *Selinger and Nelson, 1989*; *Selinger, 1991* for a theoretical account and *Drouin-Touchette et al., 2022* for recent developments). As we will detail in the following, activity profoundly alters this scenario by acting as a mechanical bandpass filter, which renders hexatic order *dominant* at length scales $\ell \ll \ell_6$ and nematic order at length scales $\ell \gg \ell_2$. We stress that by dominant, here we intend able to drive morphological features, dynamical behaviors, and fluctuations reflecting the underlying orientational order. At intermediate length scales, i.e. $\ell_6 \ll \ell \ll \ell_2$, there is no dominant order and the system's collective behavior is determined by the complex interplay of competing active and passive effects. To make progress, here we focus on the most dramatic hexatic- and nematic-dominated behaviors and treat intermediate length scales as simply as possible.

## Multiscale order in epithelia

To elucidate the multiscale organization of the system, we next compute the structure factor $S(|q|)$, using the classic framework of fluctuating hydrodynamics (see e.g. *Ramaswamy et al., 2003*). To this end, we assume both the nematic and the hexatic scalar order parameters to be uniform throughout the system and set $k_d = k_a$ and $\lambda_p = 0$ for simplicity. We stress that the validity of this approximation is strictly related with the present comparison between the hydrodynamic theory presented in this article and cell-resolved models. An assessment of the relevance of this and the other material parameters featured in *Equation 2a* can only be achieved via experimental scrutiny and is likely to depend on the specific cell type and environmental conditions. Furthermore, as the typical Reynolds number of collective epithelial flow is in the range $10^{-7}$–$10^{-6}$, we neglect inertial effects: i.e. $\rho Dv/Dt = 0$. With these simplifications, whose legitimacy will be assessed a posteriori, one can reduce *Equation 2b* to three coupled differential equations for the density and the phases of the hexatic and nematic order parameter tensors (see Methods). These equations, in turn, can be linearized about the trivial configuration, where all fields are spatially uniform and $v = 0$, and augmented with noise terms to give the following *exact* asymptotic expansion

$$S(|q|) \sim \frac{s_{-2}}{|q|^2} + s_\beta |q|^\beta \ . \tag{8}$$

The first term entails the typical giant number density fluctuations associated with the active nematic behavior at the large scale, with $s_{-2} \sim \alpha_2^2$. This effect is overestimated at the linear order, leading to an inverse quadratic dependence on the wave number $|q|$ (*Ramaswamy et al., 2003*), but is

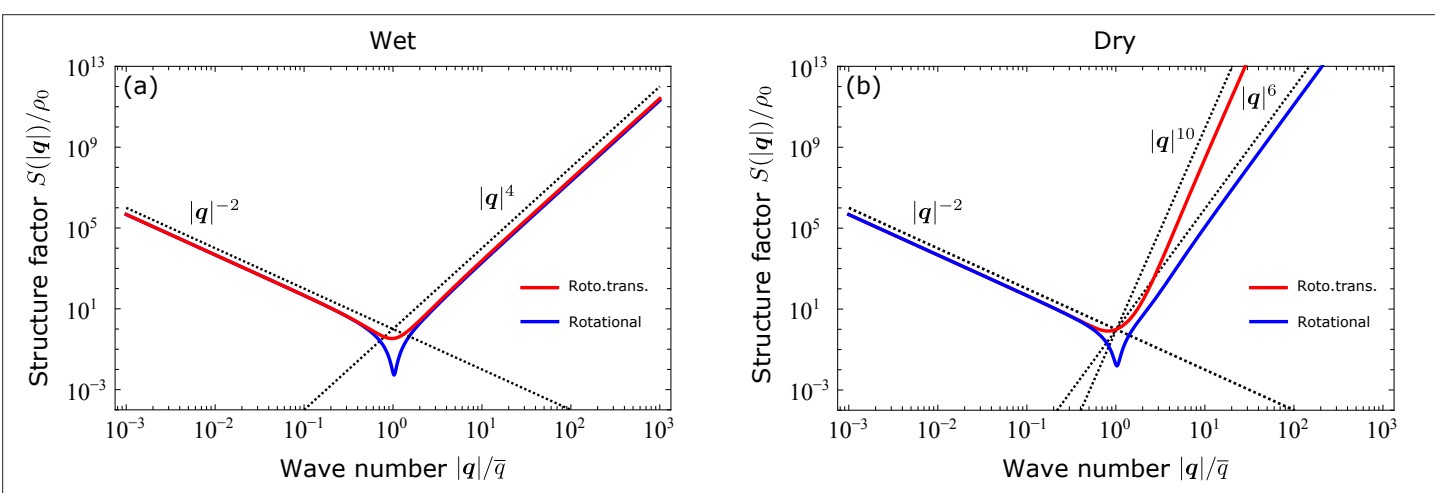

**Figure 2.** Structure factor $S(|q|)$ obtained from the analytical solutions of the linearized hydrodynamic equations in the presence of two different noise fields: purely rotational (blue) and rototranslational (red). The full analytical expression of $S(|q|)$ is given in Methods, together with a derivation of the exact asymptotic expansions of *Equation (8)*. (**a**) As long as viscous dissipation takes place (i.e. 'wet' regime), $S(|q|) \sim |q|^4$ in the limit $|q| \to \infty$, irrespective of the type of noise. (**b**) On the other hand, when friction is the sole momentum dissipation mechanism at play ('dry' regime), $S(|q|) \sim |q|^6$ in case of rotational noise and $S(|q|) \sim |q|^{10}$ when noise is both rotational and translational. In both panels, the wave number $|q|$ is rescaled by $\bar{q} = 2\pi/\bar{\ell}$, with $\bar{\ell} = (\ell_2 + \ell_6)/2$ and $\ell_2$ and $\ell_6$ as defined in *Equation (7)*.

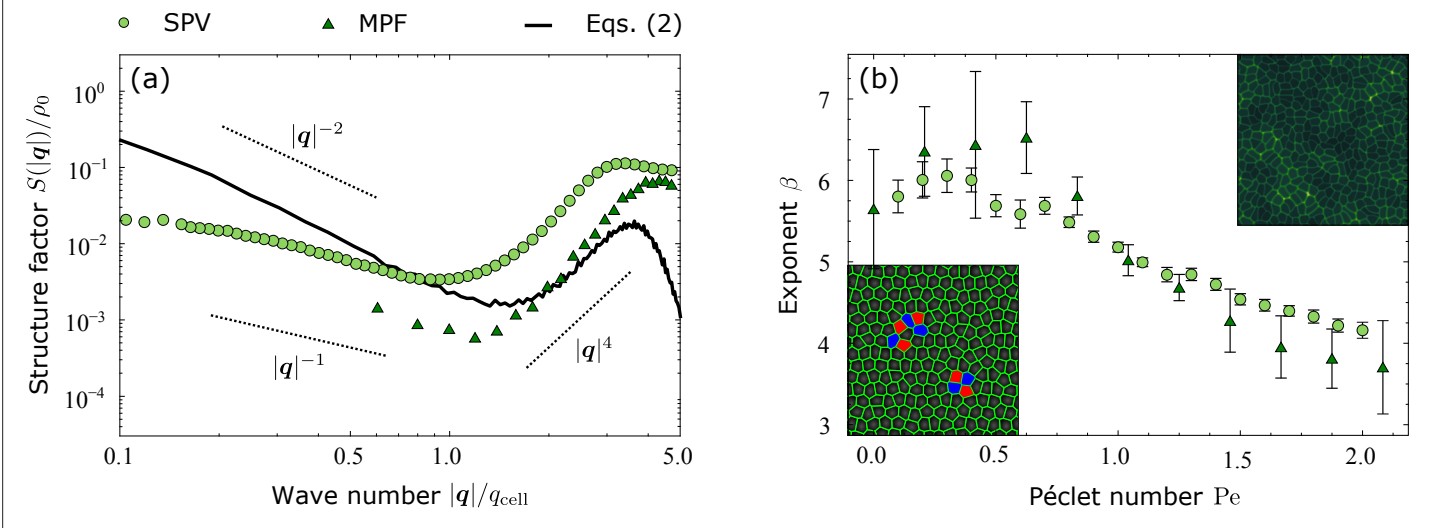

**Figure 3.** Density fluctuations in model epithelia. (**a**) Structure factor of model-epithelia calculated from a numerical integration of *Equation 2a* (black line) and from simulations of two different cell-resolved models: i.e. the self-propelled Voronoi model (SPV, red) and the multiphase field (MPF) model (blue), for a particular choice of parameters. The dashed diagonal lines mark the scaling regimes obtained analytically at the linear order, *Equation (8)*, and the wave number $|\boldsymbol{q}|$ is rescaled by $q_{\mathrm{cell}} = 2\pi/\Delta x_{\mathrm{LB}}$, where $\Delta x_{\mathrm{LB}}$ is the grid size used by the Lattice Boltzmann integrator (see Methods for details). (**b**) The exponent $\beta$, as defined in *Equation (8)*, versus the Péclet number $\mathrm{Pe}$, reflecting the persistence of directed cellular motion in front of diffusion. Error bars calculated as standard error over $n=250$ configurations for both the SPV and the MPF models. Insets: typical configurations of the SPV (bottom left) and MPF (top right) models.

The online version of this article includes the following source data for figure 3:

**Source data 1.**

generally renormalized by nonlinearities, so that $\lim_{|\boldsymbol{q}|\to 0} S(|\boldsymbol{q}|) \sim |\boldsymbol{q}|^{-\alpha}$, with $1 < \alpha < 2$ (*Shankar et al., 2018*; *Chaté, 2020*).

The second term, on the other hand, reflects the sixfold symmetry characterizing the structure of epithelia at the small length scale, with $s_\beta \sim \alpha_6^2$ and the exponent $\beta$ determined by the specific energy dissipation mechanism, as well as by the specific structure of the noise. As detailed in Methods, here we consider *four* alternative scenarios, obtained upon combining two different momentum dissipation mechanisms (i.e. viscosity and friction) with two different types of noise (i.e. rototranslational and purely rotational). In the presence of *viscous* dissipation, i.e. a regime referred to as 'wet' in the jargon of active matter, $\beta = 4$ irrespective of the nature of noise. Conversely, in the 'dry' limit, when the shear and bulk viscosity vanish and momentum dissipation solely results from the frictional interactions with the substrate, $\beta$ differs depending on whether noise affects both cells' orientational and translational dynamics, or only the former. Specifically, when only orientational noise is considered, $\beta = 6$. In contrast, $\beta = 10$ in the presence of *conservative* rototranslational noise. We again stress that *Equation (8)* is an exact asymptotic expansion, as one could verify upon comparison with the full analytical solutions plotted in *Figure 2*, and *not* a truncated power series.

To test the significance of these predictions and connect the present hydrodynamic theory with the existing literature, in *Figure 3a* we compare the structure factor obtained from numerical simulations of two different cell-resolved models of epithelia – i.e. the self-propelled Voronoi (SPV) model (*Bi et al., 2016*) and the multiphase field (MPF) model (*Loewe et al., 2020*) (see the insets *Figure 3b* for typical configurations of the two models) – with that resulting from a numerical integration of *Equation 2a* (*Carenza et al., 2019*; *Carenza et al., 2020*), with *none* of the simplifications behind *Equation (8)*. In both microscopic models, cells are treated as persistent random walkers, self-propelling at constant speed $v_0$ and whose direction of motion undergoes rotational diffusion with diffusion coefficient $D_r$ (see Methods for details). Noise is therefore expected to affect both the rotational and translational dynamics of the cell monolayer, although in a way that, unlike in our analytical treatment, cannot be trivially decoupled. Consistently with our linear analysis, both data sets exhibit two different power-law scaling regimes at small and large length scales. At small length scales, the structure factor

scales like $S(|\boldsymbol{q}|) \sim |\boldsymbol{q}|^\beta$, with $\beta$ monotonically decreasing from 6 to 4 upon increasing the Péclet number $\mathrm{Pe} = \xi_0/a$ expressing the ratio between cells' persistence length $\xi_0 = v_0/D_\mathrm{r}$ and their typical size $a$ (see **Figure 3b**).

Conversely, at large length scales, the structure factor scales like an inverse power law, with exponent consistent with the large-scale behavior of active nematics (**Chaté, 2020**). These observations can be rationalized in the light of the previous fluctuating hydrodynamic analysis. In the limit $\mathrm{Pe} \to 0$, cells do not self-propel, noise is predominantly orientational and momentum propagates only at distances comparable to the average cell size. Under this circumstances, an in silico cell layer, whether modeled via the SPV or the MPF, behaves therefore as a 'dry' active system subject to purely rotational noise, for which, consistently with our analysis, $\beta = 6$. Increasing Pe has the twofold effect of converting noise from purely rotational to rototranslational and, by stimulating cooperativity in the cellular motion, to increase the range of momentum propagation, thus driving a crossover of the cell layer from 'dry' to 'wet', hence from $\beta = 6$ to $\beta = 4$. The simple linear calculation, summarized in Methods, does not allow us to resolve the full crossover, but does provide a precise estimate of the upper and lower bounds. Finally, along the wet–dry crossover, viscosity must emerge from the cells' lateral interactions. A precise understanding of this process is outside of the scope of the present work, but recent numerical work on the Vertex model has already highlighted the existence of a rich landscape of exotic rheological phenomena, resulting from the interplay between cellular motion, morphology, and adhesion (**Tong et al., 2022**; **Hertaeg et al., 2022**). The latter could possibly explain the non-monotonic behavior at small Pe values, as a crossover from a shear-thinning to the shear-thickening behavior (**Hertaeg et al., 2022**) for additional numerical evidence of this effect.

A different signature of multiscale hexanematic order can be identified in the structure of the cross-correlation function

$$C_{26}(\boldsymbol{r}) = \frac{\langle \psi_2(\boldsymbol{r})\psi_6^*(\boldsymbol{0}) + \psi_2^*(\boldsymbol{r})\psi_6(\boldsymbol{0}) \rangle}{2} \ . \tag{9}$$

At equilibrium, and if deformations are sufficiently gentle to render backflow effects negligible, its behavior can be divided in two regimes, depending on how the distance $|\boldsymbol{r}|$ compares to the length scales $\ell_{\chi,2}$ and $\ell_{\chi,6}$ defined in the previous section and expressing the typical distance at which the mutual alignment rate of the hexatic and nematic orientations overcome that of rotational diffusion. In the simplest possible setting, when $\ell_{\chi,2} = \ell_{\chi,6} = \ell_\chi$, fluctuations dominate at short distances and the hexatic and nematic orientations are uncorrelated. Thus, $C_{26}(\boldsymbol{r})$ is approximatively constant for $|\boldsymbol{r}| \ll \ell_\chi$. The picture is reversed for $|\boldsymbol{r}| \gg \ell_\chi$. In this range, the hexatic and nematic orientations are 'locked' in a parallel configuration, i.e. $\mathrm{Arg}(\psi_2)/2 \approx \mathrm{Arg}(\psi_6)/6$, or tilted by $\pi/6$ with respect to each other, depending on the sign of the constant $\chi_{2,6}$, and the cross-correlation function exhibits the standard power-law decay characterizing two-dimensional liquid crystals with a single-order parameter: i.e. $C_{26}(\boldsymbol{r}) \sim (|\boldsymbol{r}|/\ell_\chi)^{-\eta_{26}}$, with $\eta_{26}$ a specific instance of the generic non-universal exponent $\eta_{26} = 6k_B T/(\pi K)$, with $K$ the orientational stiffness of both phases (proportional to $L_2 = L_6$). An analytical treatment of this simple case is reported in Methods. In the more generic case, in which $\ell_{\chi,2} \neq \ell_{\chi,6}$ and the relaxation rates of the hexatic and nematic phase differ, the cross-correlation function has a less standard functional form, but still features a slow and fast decay regime at short and large distances, respectively. An example of such a scenario, obtained from a numerical integration of **Equation 2a** with $\alpha_2 = 0$ and $\alpha_6 = 0$, is shown in **Figure 4a**. The curves in **Figure 4b** correspond instead to simulated configurations of the cross-correlation function of $C_{26}(\boldsymbol{r})$ for finite hexatic and nematic activity. In this case, the cross-correlation function exhibits an oscillatory behavior at short distances and vanishes at a length scale that becomes progressively large as the hexatic activity is increased. Consistently with our previous analysis, this latter feature confirms the existence of a hierarchy of orientationally ordered structures nested into each other at different length scales.

Taken together, our calculations of the structure factor and the cross-correlation function demonstrate that the hydrodynamic theory embodied in **Equations 2 and 6** is able to account for the multiscale hexanematic order observed in experiments (**Armengol-Collado et al., 2023**; **Eckert et al., 2023**) and harnesses it into a continuum mechanical framework. Whereas the origin of hexanematic order is still a matter of investigation, the current experimental and numerical evidence suggests that, similarly to granular materials (**Majmudar and Behringer, 2005**), large-scale nematic order could arise from the self-organization of the microscopic force hexapoles into *force chains*. The possibility of

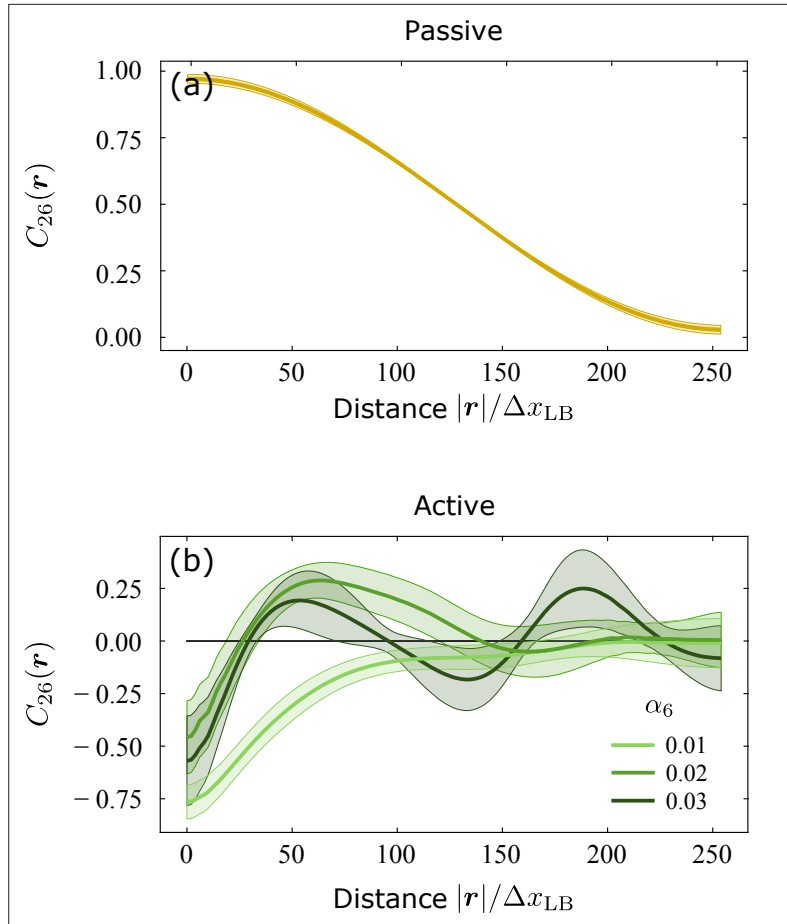

**Figure 4.** Cross-correlation function $C_{26}(\mathbf{r})$, as defined in *Equation 9*, obtained from a numerical integration of *Equation 2a* augmented with rotational noise. (**a**) In the passive case, when $\alpha_2 = 0$ and $\alpha_6 = 0$, the correlation function decays with |**r**| at a rate i.e. lower at short distances, where the dynamics of the hexatic and nematic orientations is dominated by fluctuations, and larger at long distances, where the orientations are 'locked' in a parallel configuration, or tilted by $\pi/6$ with respect to each other. (**b**) In the active case, conversely, the cross-correlation function has a damped oscillatory behavior. Consistently with *Equation (7)* and the related discussion, the range of the oscillations, corresponding to the distance at which these are fully damped, increases with the hexatic activity $\alpha_6$, indicating an enhancement of hexatic order at larger length scales. Shaded region corresponding to the standard deviation of $n = 500$ configurations. Distance is expressed in terms of the grid size $\Delta x_{\text{LB}}$ used by the Lattice Boltzmann integrator (see Methods for details).

The online version of this article includes the following source data for figure 4:

**Source data 1.**

similarity between these two phenomena has also been in relation to the initial phase of *Drosophila* gastrulation, where linear arrays of cells simultaneously undergo apical constriction in the ventral furrow region (*Jason Gao et al., 2016*).

## Conclusions

In conclusion, we have introduced a continuum model of collectively migrating layers of epithelial cells, built upon a recent generalization of the hydrodynamic theory of *p*-atic liquid crystals (*Giomi et al., 2022a*; *Giomi et al., 2022b*). This approach allows one to account for arbitrary discrete rotational symmetries, thereby going beyond existing hydrodynamic theories of epithelia (*Ranft et al., 2010*; *Popović et al., 2017*; *Pérez-González et al., 2019*; *Ishihara et al., 2017*; *Czajkowski et al., 2018*; *Hernandez and Marchetti, 2021*; *Grossman and Joanny, 2022*), where the algebraic structure of the hydrodynamic variables renders impossible to account for liquid crystal order other than isotropic (i.e. $p = 0$), polar (i.e. $p = 1$), or nematic (i.e. $p = 2$). Upon computing the static structure factor and

comparing this with the outcome of two different cell-resolved models – i.e. the SPV (*Bi et al., 2016*) and MPF (*Loewe et al., 2020*) models – we have shown that, consistently with recent experimental findings (*Armengol-Collado et al., 2023*; *Eckert et al., 2023*), epithelial layers may in fact comprise both nematic and hexatic (i.e. $p = 6$) order, coexisting at different length scales. Although the consequences of such a remarkable versatility are yet to be explored, we expect hexatic order to be relevant for short-scale remodeling events, where the local nature of hexatic order, combined with the rich dynamics of hexatic defects (*Zippelius et al., 1980*; *Amir and Nelson, 2012*), may mediate processes such as cell intercalations and the rearrangement of multicellular *rosettes* (*Blankenship et al., 2006*; *Rauzi, 2020*). Such a local motion, in turn, may be coordinated at the large scale by the underlying nematic order, giving rise to a persistent unidirectional flow, such as that observed during wound healing and cancer progression (*Friedl and Gilmour, 2009*). Furthermore, the existence of multiscale liquid crystal order echoes the most recent understanding of phenotypic plasticity in tissues, according to which the epithelial (i.e. solid-like) and mesenchymal (i.e. liquid-like) states represent the two ends of a spectrum of intermediate phenotypes (*Zhang and Weinberg, 2018*). These intermediate states display distinctive cellular characteristics, including adhesion, motility, stemness and, in the case of cancer cells, invasiveness, drug resistance, etc. Can multiscale liquid crystal order help understanding how the biophysical properties of tissues vary along the epithelial–mesenchymal spectrum? This and related questions will be addressed in the near future.

## Methods

### Quantification of *p*-atic order in epithelial layers

Following *Armengol-Collado et al., 2023*, we use the *shape function* $\gamma_p$ to quantify the amount of $p$-fold symmetry of an arbitrary cell. Denoting $\boldsymbol{r}_v$ with $v = 1, 2 \ldots V$, the positions of its vertices with respect to the cell's center of mass, one has

$$\gamma_p = \frac{\sum_{v=1}^{V} |\boldsymbol{r}_v|^p e^{ip\phi_v}}{\sum_{v=1}^{V} |\boldsymbol{r}_v|^p} , \tag{10}$$

with $\phi_v = \mathrm{Arg}(\boldsymbol{r}_v)$ the angle between $\boldsymbol{r}_v$ and the $x$-axis of a Cartesian frame. A schematic representation of these elements in an arbitrary irregular polygon is shown in *Figure 5a*. Unlike the complex function $\psi_p = e^{ip\vartheta}$, which has unit magnitude by construction, the magnitude $|\gamma_p|$ quantify the resemblance of a generic polygon with a *regular p*-sided polygon of the same size, while the phase $\vartheta = \mathrm{Arg}(\gamma_p)/p$ marks the orientation of the polygon. For regular V-sided polygons, $|\gamma_p| = 1$ provided $p$ is an integer multiple of $V$ and $|\gamma_p| \approx 0$ otherwise. Furthermore, from $\gamma_p$ one can readily compute.

$$\psi_p = \frac{\gamma_p}{|\gamma_p|}. \tag{11}$$

*Figure 5b, c* shows examples of the functions $\gamma_2$ and $\gamma_6$ for a typical configuration of the SPV. We emphasize that $\gamma_p$, which, as shown in *Armengol-Collado et al., 2023*, arises from a $p$-fold generalization of the classic shape tensor (*Aubouy et al., 2003*), is solely determined by the positions of the vertices of an individual polygon and, therefore, does not depend on the spatial organization of the neighboring cells. As a consequence, this approach establishes an orientation purely based on cellular *shape*, thereby eliminating the arbitrariness involved with associating a network of bonds to a planar tessellation, where the latter is not inherent.

The shape function $\gamma_p$ can then be coarse grained at the length scale $\ell$ to construct the *shape parameter*:

$$\Gamma_p(\boldsymbol{r}) = \frac{1}{N_\ell} \sum_{c=1} \gamma_p(\boldsymbol{r}_c)\, \Theta(\ell - |\boldsymbol{r} - \boldsymbol{r}_c|) , \tag{12}$$

where the $\boldsymbol{r}_c$ is the position of the $c$th cell, $\Theta$ is the Heaviside step function, such that $\Theta(x) = 1$ for $x > 0$ and 0 otherwise, and $N_\ell = \sum_c \Theta(\ell - |\boldsymbol{r} - \boldsymbol{r}_c|)$ is the number of cells within a distance $\ell$ from $\boldsymbol{r}_c$. As in the case of $\gamma_p$, the magnitude of $\Gamma_p$ reflects the resemblance between a multicelluar cluster and a regular $p$-sided polygon, while its phase marks the cluster's global orientation. The outcome of an application of this method to the Voronoi model is illustrated in *Figure 6* for $p = 2$. The different patches in panel

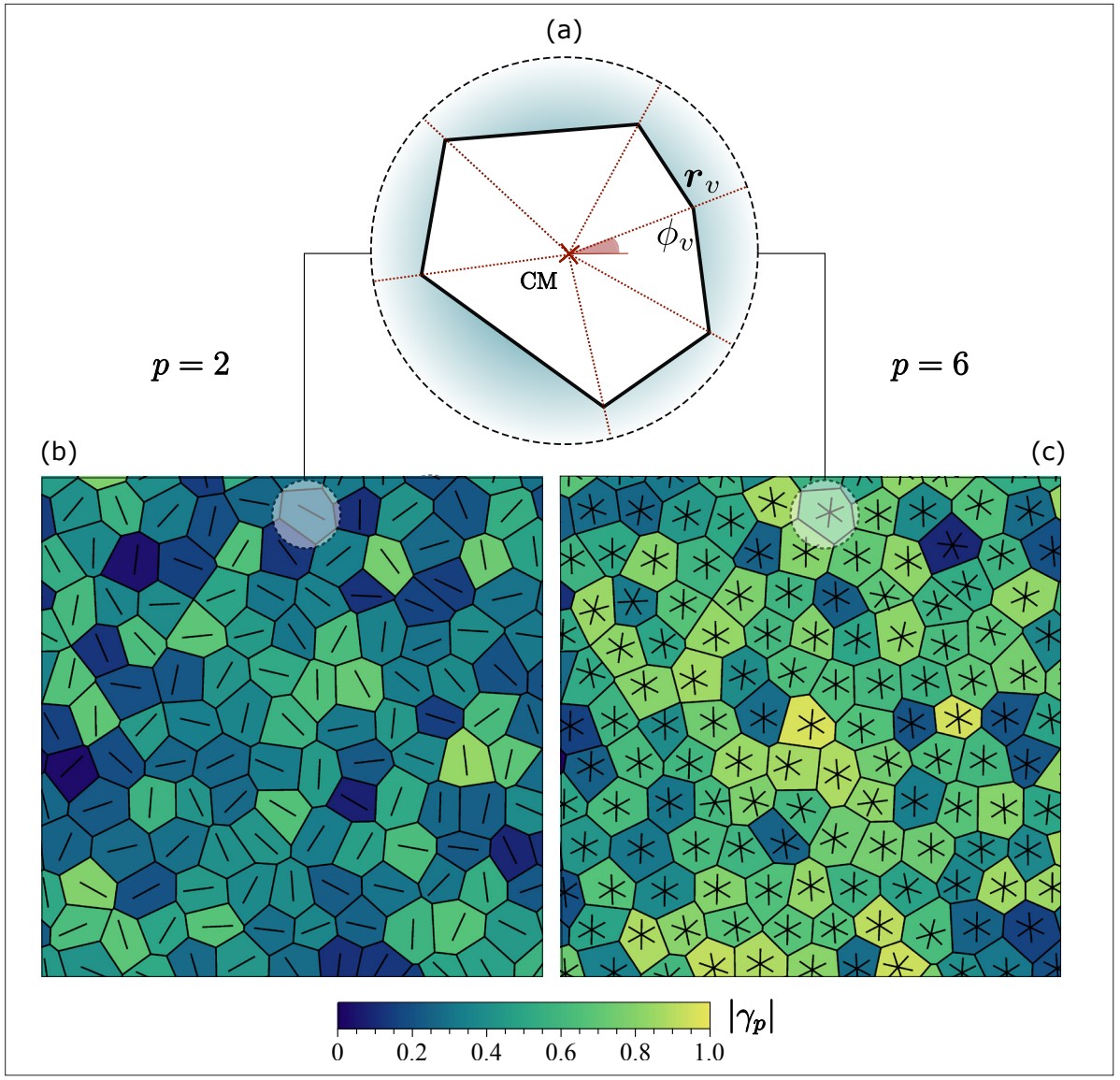

**Figure 5.** Nematic and hexatic shape function. (**a**) Irregular polygonal cell with a red cross marking its center of mass and $\boldsymbol{r}_v$ and $\phi_v$ the radial vector and the angle to one of the six vertices, respectively. (**b**) and (**c**) show the same tessellation of the plane with cells of different shapes and the shape analysis using the function in *Equation 10* for the nematic ($p = 2$) and hexatic ($p = 6$) case. Rods and stars are oriented according to the phase of $\gamma_p$ and the color corresponds to its magnitude.

(a) are regions with uniform $\theta = \mathrm{Arg}(\Gamma_2)/2$, while in panel (b), there are plotted streamlines showing the orientation of the director $\boldsymbol{n} = \cos\theta\,\boldsymbol{e}_x + \sin\theta\,\boldsymbol{e}_y$.

## Passive stresses

As explained in the main text, the passive contribution to the stress tensor is given by $\boldsymbol{\sigma}^{(\mathrm{p})} = -P\mathbb{1} + \boldsymbol{\sigma}^{(\mathrm{e})} + \boldsymbol{\sigma}^{(\mathrm{r})} + \boldsymbol{\sigma}^{(\mathrm{v})}$, where, as demonstrated in *Giomi et al., 2022b*

$$\sigma_{ij}^{(\mathrm{e})} = -L_p \partial_i \boldsymbol{Q}_p \odot \partial_j \boldsymbol{Q}_p \,, \tag{13a}$$

$$\sigma_{ij}^{(\mathrm{r})} = -\bar{\lambda}_p \boldsymbol{Q}_p \odot \boldsymbol{H}_p \, \delta_{ij} + (-1)^{p-1} \lambda_p \partial_{k_1 k_2 \cdots k_{p-2}}^{p-2} H_{k_1 k_2 \cdots ij} + \frac{p}{2} \left( Q_{k_1 k_2 \cdots i} H_{k_1 k_2 \cdots j} - H_{k_1 k_2 \cdots i} Q_{k_1 k_2 \cdots j} \right) \,, \tag{13b}$$

$$\sigma_{ij}^{(\mathrm{v})} = 2\eta \left[ u_{ij} \right] + \zeta \, \mathrm{tr}(\mathbf{u}) \, \delta_{ij} \,, \tag{13c}$$

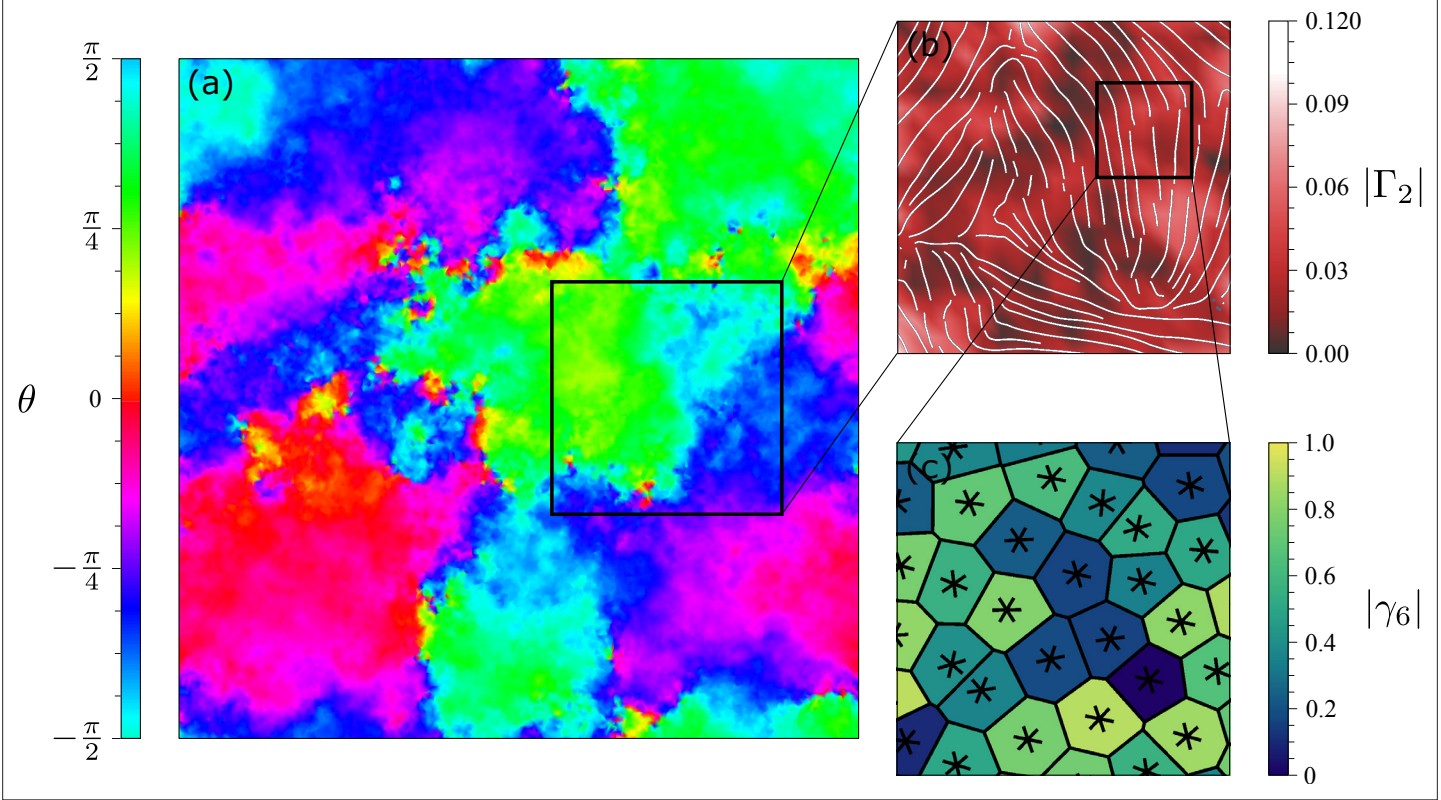

**Figure 6.** Hexanematic symmetry of Voronoi model. (**a**) Coarse-grained nematic orientation $\theta$ obtained from averaging the local shape of cells over domains of size $30\,\ell_{\text{cell}}$, with $\ell_{\text{cell}}$ the average size of individual cells. Regions with the same color represent domains of coherent nematic orientation. (**b**) Part of the system where we use $\Gamma_2$ to characterize the nematic phase. Solid lines represent the nematic director and the color indticates the magnitude of the nematic shape function. (**c**) Voronoi cell structure of a region where the nematic field is uniform. Polygons are colored according to $|\gamma_6|$ and the stars are oriented according to $\text{Arg}(\gamma_6)/6$.

where $\eta$ and $\zeta$ are, respectively, the shear and bulk viscosity and the other material parameters are defined in the main text. Under the assumptions of uniform order parameter, i.e. $|\mathbf{Q}_p|^2 = |\Psi_p|^2/2 = \text{const}$, and taking $\lambda_p = 0$, **Equation 13a** reduces to the expression derived in **Zippelius, 1980**; **Zippelius et al., 1980**. That is

$$\boldsymbol{\sigma}^{(e)} + \boldsymbol{\sigma}^{(r)} = -P\mathbb{1} + \frac{K_p}{2}\,\boldsymbol{\varepsilon}\nabla^2\theta - K_p\nabla\theta \otimes \nabla\theta\,, \tag{14}$$

where the first term in **Equation 13b** has incorporated into the pressure $P$ and $K_p$ denotes the orientational stiffness of the $p$-atic phase, related to the order parameter stiffness by

$$K_p = \frac{p^2|\Psi_p|^2}{2}\,L_p \tag{15}$$

and $\boldsymbol{\varepsilon}$ is the two-dimensional antisymmetric tensor, with $\varepsilon_{xy} = -\varepsilon_{yx} = 1$ and $\varepsilon_{xx} = \varepsilon_{yy} = 0$.

## Linear fluctuating hydrodynamics

To compute the structure factor, we follow **Ramaswamy et al., 2003** and augment **Equation 2b**, **Equation 2c** with short-ranged correlated noise field. Then calling $\vartheta$ and $\varphi$ the nematic and hexatic *fluctuating* orientation fields and linearizing the hydrodynamic equations about the homogeneous and stationary solutions, $\vartheta = \varphi = 0$ and $\boldsymbol{v} = \boldsymbol{0}$, gives

$$\partial_t\delta\rho = -\rho_0\nabla \cdot \delta\boldsymbol{v}\,, \tag{16a}$$

$$\partial_t\delta\vartheta = \mathcal{D}_2\nabla^2\delta\vartheta + \frac{1}{2}\,\boldsymbol{e}_z \cdot (\nabla \times \delta\boldsymbol{v}) + \frac{9}{4}\chi_2(\delta\vartheta - \delta\varphi) + \xi^{(\vartheta)}\,, \tag{16b}$$

$$\partial_t \delta\varphi = \mathcal{D}_6 \nabla^2 \delta\varphi + \frac{1}{2} \boldsymbol{e}_z \cdot (\nabla \times \delta\boldsymbol{v}) + \frac{1}{4} \chi_6 (\delta\varphi - \delta\vartheta) + \xi^{(\varphi)} \, , \tag{16c}$$

where $\delta\vartheta$, $\delta\varphi$, and $\delta\boldsymbol{v}$ indicate a small departure from the homogeneous and stationary configurations of the fields $\vartheta$, $\varphi$, and $\boldsymbol{v}$, $\mathcal{D}_p = \Gamma_p L_p$, $\chi_p = \Gamma_p \chi_{2,6}$, and $\xi^{(\vartheta)}$ and $\xi^{(\varphi)}$ are short-ranged correlated noise fields: i.e.

$$\left\langle \xi^{(\alpha)}(\boldsymbol{r}, t)\xi^{(\beta)}(\boldsymbol{r}', t') \right\rangle = 2 \left( \Xi^{(\vartheta)} \delta_{\alpha\vartheta} \delta_{\beta\vartheta} + \Xi^{(\varphi)} \delta_{\alpha\varphi} \delta_{\beta\varphi} \right) \delta(\boldsymbol{r} - \boldsymbol{r}')\delta(t - t') \, . \tag{17}$$

The velocity field $\delta\boldsymbol{v}$, on the other hand, is found from the Stokes limit of *Equation 2b* in the main text, which, at the linear order in all fluctuating fields, takes the form

$$\eta\nabla^2\delta\boldsymbol{v} + \zeta\nabla(\nabla \cdot \delta\boldsymbol{v}) - \varsigma\delta\boldsymbol{v} + \boldsymbol{f}^{(\mathrm{p})} + \boldsymbol{f}^{(\mathrm{a})} + \boldsymbol{\xi}^{(v)} = \boldsymbol{0} \, . \tag{18}$$

where $\boldsymbol{f}^{(\mathrm{p})} = \nabla \cdot \boldsymbol{\sigma}^{(\mathrm{p})}$ and $\boldsymbol{f}^{(\mathrm{a})} = \nabla \cdot \boldsymbol{\sigma}^{(\mathrm{a})}$ are the body forces resulting from the passive and active stresses, respectively. The quantity $\boldsymbol{\xi}^{(v)}$ is a translational noise field. In the absence of external stimuli, it is reasonable to assume that global momentum is neither created nor dissipated by translational fluctuations, but only redistributed across the cell layer. Thus $\boldsymbol{\xi}^{(v)}$ is either conservative or null, from which

$$\left\langle \xi_i^{(v)}(\boldsymbol{r}, t)\xi_j^{(v)}(\boldsymbol{r}', t') \right\rangle = 2 \Xi^{(v)} \delta_{ij}(-\nabla^2)\delta(\boldsymbol{r} - \boldsymbol{r}')\delta(t - t') \, , \tag{19}$$

with $\{i, j\} \in \{x, y\}$ and the case of noiseless translational dynamics, corresponding to *Figure 3* in the main text, is recovered in the limit $\Xi^{(v)} \to 0$. The pressure $P$, in turn, can be related to the density by a linear equation of state of the form

$$P = c_{\mathrm{s}}^2 \rho \, , \tag{20}$$

with $c_{\mathrm{s}}$ the speed of sound. Together with the expression for the active stress given in *Equation 3* of the main text, this gives

$$\boldsymbol{f}^{(\mathrm{p})} = \left( -c_{\mathrm{s}}^2 \partial_x \delta\rho + \frac{K_2}{2} \partial_y \nabla^2 \delta\vartheta + \frac{K_6}{2} \partial_y \nabla^2 \delta\varphi \right) \boldsymbol{e}_x - \left( c_{\mathrm{s}}^2 \partial_y \delta\rho + \frac{K_2}{2} \partial_x \nabla^2 \delta\vartheta + \frac{K_6}{2} \partial_x \nabla^2 \delta\varphi \right) \boldsymbol{e}_y \, , \tag{21a}$$

$$\boldsymbol{f}^{(\mathrm{a})} = \left[ \alpha_2 \partial_y \delta\vartheta + \frac{3}{2} \alpha_6 \left( \partial_y^4 - 5\partial_x^2\partial_y^2 + \frac{5}{2} \partial_x^4 \right) \partial_y \delta\varphi \right] \boldsymbol{e}_x + \left[ \alpha_2 \partial_x \delta\vartheta + \frac{3}{2} \alpha_6 \left( \partial_x^4 - 5\partial_x^2\partial_y^2 + \frac{5}{2} \partial_y^4 \right) \partial_x \delta\varphi \right] \boldsymbol{e}_y \tag{21b}$$

Now, in Fourier space *Equation 18* can be cast in the form of the following linear algebraic equation

$$\left[ \left( \eta|\boldsymbol{q}|^2 + \varsigma \right) \mathbb{1} + \zeta\boldsymbol{q} \otimes \boldsymbol{q} \right] \cdot \delta\hat{\boldsymbol{v}} = \hat{\boldsymbol{f}}^{(\mathrm{p})} + \hat{\boldsymbol{f}}^{(\mathrm{a})} + \hat{\boldsymbol{\xi}}^{(v)} \, , \tag{22}$$

where the hat denotes Fourier transformation. Next, using

$$\left[ \left( \eta|\boldsymbol{q}|^2 + \varsigma \right) \mathbb{1} + \zeta\boldsymbol{q} \otimes \boldsymbol{q} \right]^{-1} = \frac{\left[ (\eta + \zeta)|\boldsymbol{q}|^2 + \varsigma \right] \mathbb{1} - \zeta\boldsymbol{q} \otimes \boldsymbol{q}}{(\eta|\boldsymbol{q}|^2 + \varsigma)[(\eta + \zeta)|\boldsymbol{q}|^2 + \varsigma]} \, , \tag{23}$$

and solving *Equation 22* and incorporating the resulting velocity field in *Equation 16a* gives, after several algebraic manipulation

$$-i\omega \begin{bmatrix} \delta\hat{\rho} \\ \delta\hat{\vartheta} \\ \delta\hat{\varphi} \end{bmatrix} = \hat{\boldsymbol{M}} \cdot \begin{bmatrix} \delta\hat{\rho} \\ \delta\hat{\vartheta} \\ \delta\hat{\varphi} \end{bmatrix} + \begin{bmatrix} \hat{\eta}^{(\rho)} \\ \hat{\eta}^{(\vartheta)} \\ \hat{\eta}^{(\varphi)} \end{bmatrix} \, , \tag{24}$$

where the matrix $\hat{\boldsymbol{M}}$ is given by

$$\hat{M} = \begin{bmatrix} -\dfrac{\rho_0 c_s^2 |q|^2}{(\eta+\zeta)|q|^2+\varsigma} & \dfrac{2\rho_0\alpha_2 q_x q_y}{(\eta+\zeta)|q|^2+\varsigma} & \dfrac{3\rho_0\alpha_6\left(3q_x^5 q_y - 10q_x^3 q_y^3 + 3q_x q_y^5\right)}{2\left[(\eta+\zeta)|q|^2+\varsigma\right]} \\ 0 & -\mathcal{D}_2|q|^2 - \dfrac{K_2|q|^4}{4(\eta|q|^2)+\varsigma} + \dfrac{9}{4}\chi_2 - \dfrac{\alpha_2\left(q_x^2-q_y^2\right)}{(\eta)|q|^2+\varsigma} & -\dfrac{K_6|q|^4}{4(\eta)|q|^2+\varsigma} - \dfrac{9}{4}\chi_2 - \dfrac{3\alpha_6\left(q_x^6 - 15q_x^4 q_y^2 + 15q_x^2 q_y^4 - q_y^6\right)}{8(\eta)|q|^2+\varsigma)} \\ 0 & -\dfrac{K_2|q|^4}{4(\eta)|q|^2+\varsigma} - \dfrac{1}{4}\chi_6 - \dfrac{\alpha_2\left(q_x^2-q_y^2\right)}{2(\eta)|q|^2+\varsigma)} & -\mathcal{D}_6|q|^2 - \dfrac{K_6|q|^4}{4(\eta)|q|^2+\varsigma} + \dfrac{1}{4}\chi_6 - \dfrac{3\alpha_6\left(q_x^6 - 15q_x^4 q_y^2 + 15q_x^2 q_y^4 - q_y^6\right)}{8(\eta)|q|^2+\varsigma} \end{bmatrix}$$

and the functions $\eta^{(\alpha)}$, with $\alpha \in \{\rho, \vartheta, \varphi\}$, are effective noise fields whose correlation functions are given by

$$\left\langle \hat{\eta}^{(\alpha)}(q,\omega)\hat{\eta}^{(\beta)}(q',\omega')\right\rangle = (2\pi)^3\, 2\hat{H}^{(\alpha)}(q)\delta_{\alpha\beta}\delta(q+q')\delta(\omega+\omega')\,, \tag{25}$$

where the functions $\hat{H}^{(\alpha)} = \hat{H}^{(\alpha)}(q)$ are given by

$$\hat{H}^{(\rho)} = \frac{\rho_0^2|q|^4}{[(\eta+\zeta)|q|^2+\varsigma]^2}\,\Xi^{(v)}\,, \tag{26}$$

$$\hat{H}^{(\alpha)} = \Xi^{(\vartheta)}\delta_{\alpha\vartheta} + \Xi^{(\varphi)}\delta_{\alpha\varphi} + \frac{|q|^4}{4(\eta|q|^2+\varsigma)^2}\,\Xi^{(v)}\,. \tag{27}$$

Notice that, while hydrodynamic flow has the effect of coloring the orientational noise embodied in the stochastic fields $\xi^{(\vartheta)}$ and $\xi^{(\varphi)}$, via the vorticity field on the right-hand side of *Equation 16b*, *Equation 16c*, this effect disappears at the small (i.e. $|q| \to \infty$) and large (i.e. $|q| \to 0$) scale, as long as *both* viscous and frictional dissipation are present.

## Structure factor

The static structure factor can be expressed in integral form as

$$S(q) = \int_{-\infty}^{\infty}\frac{d\omega}{2\pi}\,S(q,\omega)\,. \tag{28}$$

where the dynamic structure factor $S(q,\omega)$, can be calculated from the correlation function

$$\left\langle \delta\hat{\rho}(q,\omega)\delta\hat{\rho}(q',\omega')\right\rangle = (2\pi)^3 S(q,\omega)\delta(q+q')\delta(\omega+\omega')\,. \tag{29}$$

To compute the left-hand side of *Equation 29* one can solve *Equation 24* with respect to $\delta\hat{\rho}$, $\delta\hat{\vartheta}$, and $\delta\hat{\varphi}$. This gives

$$\delta\hat{\rho} = \frac{i\hat{\eta}^{(\rho)}}{\omega - i\hat{M}_{\rho\rho}} - \frac{\hat{\eta}^{(\vartheta)}\left[\hat{M}_{\rho\vartheta}\left(\omega - i\hat{M}_{\varphi\varphi}\right) + i\hat{M}_{\rho\varphi}\hat{M}_{\varphi\vartheta}\right] + \hat{\eta}^{(\varphi)}\left[\hat{M}_{\rho\varphi}\left(\omega - i\hat{M}_{\vartheta\vartheta}\right) + i\hat{M}_{\rho\vartheta}\hat{M}_{\vartheta\varphi}\right]}{\left(\omega - i\hat{M}_{\rho\rho}\right)\left[\omega^2 - i\omega(\hat{M}_{\vartheta\vartheta} + \hat{M}_{\varphi\varphi}) - \hat{M}_{\vartheta\vartheta}\hat{M}_{\varphi\varphi} + \hat{M}_{\vartheta\varphi}\hat{M}_{\varphi\vartheta}\right]}\,, \tag{30a}$$

$$\delta\hat{\vartheta} = \frac{\hat{\eta}^{(\vartheta)}(i\omega + \hat{M}_{\varphi\varphi}) - \hat{\eta}^{(\varphi)}\hat{M}_{\vartheta\varphi}}{\left[\omega^2 - i\omega(\hat{M}_{\vartheta\vartheta} + \hat{M}_{\varphi\varphi}) - \hat{M}_{\vartheta\vartheta}\hat{M}_{\varphi\varphi} + \hat{M}_{\vartheta\varphi}\hat{M}_{\varphi\vartheta}\right]}\,, \tag{30b}$$

$$\delta\hat{\varphi} = \frac{\hat{\eta}^{(\varphi)}(i\omega + \hat{M}_{\vartheta\vartheta}) - \hat{\eta}^{(\vartheta)}\hat{M}_{\varphi\vartheta}}{\left[\omega^2 - i\omega(\hat{M}_{\vartheta\vartheta} + \hat{M}_{\varphi\varphi}) - \hat{M}_{\vartheta\vartheta}\hat{M}_{\varphi\varphi} + \hat{M}_{\vartheta\varphi}\hat{M}_{\varphi\vartheta}\right]}\,. \tag{30c}$$

The static structure factor can then be expressed as

$$S = S^{(\rho)} + S^{(\vartheta)} + S^{(\varphi)}\,. \tag{31}$$

The first term on the right-hand side can be readily calculated in the form

$$S^{(\rho)} = \int_{-\infty}^{\infty}\frac{d\omega}{\pi}\frac{\hat{H}^{(\rho)}}{\hat{M}_{\rho\rho}^2 + \omega^2} = \frac{\hat{H}^{(\rho)}}{|\hat{M}_{\rho\rho}|} = \frac{\rho_0|q|^2\,\Xi^{(v)}}{c_s^2[(\eta+\zeta)|q|^2+\varsigma]}\,, \tag{32}$$

indicating that, if driven solely by pressure fluctuations, the system would relax toward a structureless homogeneous state with $S \to \rho_0\Xi^{(\rho)}/(\varsigma c_s^2)$ when $|q| \to 0$. The effect of the active currents is instead

accounted for by the second and third terms on the right-hand side of *Equation 31*, which can be cast in the general form

$$S^{(\alpha)} = H^{(\alpha)} \int_{-\infty}^{\infty} \frac{d\omega}{\pi} \frac{g^{(\alpha)}(\omega)}{|h(\omega)|^2} \,, \qquad \alpha = \{\vartheta, \varphi\} \,, \tag{33}$$

where

$$g^{(\vartheta)}(\omega) = (\hat{M}_{\rho\vartheta}\omega)^2 + (\hat{M}_{\rho\varphi}\hat{M}_{\varphi\vartheta} - \hat{M}_{\rho\vartheta}\hat{M}_{\varphi\varphi})^2 \,, \tag{34a}$$

$$g^{(\varphi)}(\omega) = (\hat{M}_{\rho\varphi}\omega)^2 + (\hat{M}_{\rho\vartheta}\hat{M}_{\vartheta\varphi} - \hat{M}_{\rho\varphi}\hat{M}_{\vartheta\vartheta})^2 \,, \tag{34b}$$

$$h(\omega) = \left(\omega - i\hat{M}_{\rho\rho}\right)\left[\omega^2 - i\omega(\hat{M}_{\vartheta\vartheta} + \hat{M}_{\varphi\varphi}) - \hat{M}_{\vartheta\vartheta}\hat{M}_{\varphi\varphi} + \hat{M}_{\vartheta\varphi}\hat{M}_{\varphi\vartheta}\right] \,. \tag{34c}$$

The integral over $\omega$ can be derived using the residue theorem upon computing the roots of the complex third-order polynomial $h$. To make progress, we express

$$|h(\omega)|^2 = (\omega^2 + \omega_1^2)(\omega^2 + \omega_2^2)(\omega^2 + \omega_3^2) \,, \tag{35}$$

where $\omega_1$, $\omega_2$, and $\omega_3$ are given by

$$\omega_1 = \hat{M}_{\rho\rho} \,, \tag{36a}$$

$$\omega_2 = \frac{1}{2}\left(\hat{M}_{\vartheta\vartheta} + \hat{M}_{\varphi\varphi} - \sqrt{(\hat{M}_{\vartheta\vartheta} - \hat{M}_{\varphi\varphi})^2 + 4\hat{M}_{\vartheta\varphi}\hat{M}_{\varphi\vartheta}}\right) \,, \tag{36b}$$

$$\omega_3 = \frac{1}{2}\left(\hat{M}_{\vartheta\vartheta} + \hat{M}_{\varphi\varphi} + \sqrt{(\hat{M}_{\vartheta\vartheta} - \hat{M}_{\varphi\varphi})^2 + 4\hat{M}_{\vartheta\varphi}\hat{M}_{\varphi\vartheta}}\right) \,. \tag{36c}$$

The integrand on the right-hand side of *Equation 33* has, therefore, three pairs of purely imaginary poles: i.e. $\pm i|\omega_1|$, $\pm i|\omega_2|$, and $\pm i|\omega_3|$. Next, turning the integration range to an infinite semicircular contour on the complex upper half-plane and summing the associated residues gives, after lengthy algebraic manipulations

$$S^{(\vartheta)} = \frac{H^{(\vartheta)}\left[\Omega_1\hat{M}_{\rho\vartheta}^2 + \Omega_2(\hat{M}_{\rho\varphi}\hat{M}_{\varphi\vartheta} - \hat{M}_{\rho\vartheta}\hat{M}_{\varphi\varphi})^2\right]}{\Omega_1\Omega_2\Omega_3 - \Omega_1^2} \,, \tag{37a}$$

$$S^{(\varphi)} = \frac{H^{(\varphi)}\left[\Omega_1\hat{M}_{\rho\varphi}^2 + \Omega_2(\hat{M}_{\rho\vartheta}\hat{M}_{\vartheta\varphi} - \hat{M}_{\rho\varphi}\hat{M}_{\vartheta\vartheta})^2\right]}{\Omega_1\Omega_2\Omega_3 - \Omega_1^2} \,, \tag{37b}$$

where we have set

$$\Omega_1 = |\omega_1||\omega_2||\omega_3| \,, \tag{38a}$$

$$\Omega_2 = |\omega_1| + |\omega_2| + |\omega_3| \,, \tag{38b}$$

$$\Omega_2 = |\omega_1||\omega_2| + |\omega_1||\omega_3| + |\omega_2||\omega_3| \,. \tag{38c}$$

Now, although the individual elements of the matrix $\hat{M}$ depend on the individual components of the wave vector – i.e. $q_x$ and $q_y$ – this is an artefact of linearizing the hydrodynamic equations about a specific orientation (i.e. $\vartheta = \varphi = 0$ in this case). Because of the lack of long-ranged order and of specific directions that could affect the spectrum of density fluctuations, the latter is expected to be isotropic, thus $S = S(|q|)$. To remove the fictitious angular dependence, one can either linearize *Equation 2a* about a generic pair of angles, $\vartheta_0$ and $\varphi_0$, and then use these to calculate a circular average – i.e. $S(|q|) = 1/(2\pi)^2 \int d\vartheta_0\, d\varphi_0\, S(q)$ – or, more simply, by orienting $q$ so to cancel the directional dependence. Thus, taking $q_x = q_y = |q|/\sqrt{2}$ gives a simpler expression of the matrix $\hat{M}$. That is

$$\hat{M} = \begin{bmatrix} -\dfrac{\rho_0 c_s^2 |q|^2}{(\eta+\zeta)|q|^2 + \varsigma} & \dfrac{\rho_0 \alpha_2 |q|^2}{(\eta+\zeta)|q|^2 + \varsigma} & -\dfrac{3\rho_0 \alpha_6 |q|^6}{4[(\eta+\zeta)|q|^2 + \varsigma]} \\[2ex] 0 & -\mathcal{D}_2|q|^2 - \dfrac{K_2|q|^4}{4(\eta|q|^2 + \varsigma)} + \dfrac{9}{4}\chi_2 & -\dfrac{K_6|q|^4}{4(\eta|q|^2 + \varsigma)} - \dfrac{9}{4}\chi_2 \\[2ex] 0 & -\dfrac{K_2|q|^4}{4(\eta|q|^2 + \varsigma)} - \dfrac{1}{4}\chi_6 & -\mathcal{D}_6|q|^2 - \dfrac{K_6|q|^4}{4(\eta|q|^2 + \varsigma)} + \dfrac{1}{4}\chi_6 \end{bmatrix} \,. \tag{39}$$

Using the elements of this matrix in combination with **Equations 31; 33**, **Equations 36a; 38a** yields the curves plotted in **Figure 3**. Finally, asymptotically expanding **Equation 31** allows one, after lengthy algebraic manipulations, to calculate the coefficients $s_{-2}$ and $s_4$ in **Equation 8**. That is

$$s_{-2} = \frac{\rho_0 \alpha_2^2 \left[ (9\chi_2)^2 \, \Xi_\varphi + \chi_6^2 \, \Xi_\vartheta \right]}{c_s^2 (9\chi_2 \mathcal{D}_6 + \chi_6 \mathcal{D}_2) \left[ \rho_0 c_s^2 (9\chi_2 + \chi_6) + \varsigma (9\chi_2 \mathcal{D}_6 + \chi_6 \mathcal{D}_2) \right]} \,, \tag{40a}$$

$$s_4 = \frac{72 \rho_0 \alpha_6^2 \left[ (K_2^2 + 8\eta \mathcal{D}_2 K_2 + 8\eta^2 \mathcal{D}_2^2) \, \Xi^{(v)} + K_2^2 \, \Xi_\vartheta + 2\eta^2 (K_2 + 4\eta \mathcal{D}_2)^2 \, \Xi_\varphi \right]}{c_s^2 (\eta + \zeta) \left[ K_2 + K_6 + 4\eta (\mathcal{D}_2 + \mathcal{D}_6) \right]^4} \,. \tag{40b}$$

Notice that, while both orientational and translation noise affect the amplitude of density fluctuations at small length scales, where $S(|\boldsymbol{q}|) \sim s_4 |\boldsymbol{q}|^4$, translational noise becomes unimportant at the large scale, where $S(|\boldsymbol{q}|) \sim s_{-2}/|\boldsymbol{q}|^2$. Furthermore, as long as viscous dissipation is at play, switching off translational noise (i.e. $\Xi^{(v)} \to 0$) does not alter the scaling behavior of the structure factor at neither range of length scales. Taking the dry limit (i.e. $\eta \to 0$ and $\zeta \to 0$) leaves the large-scale behavior unaltered, but does affect the scaling of density fluctuations at short length scales, where translational fluctuations are most prominent. Specifically, $S(|\boldsymbol{q}|) \sim s_6 |\boldsymbol{q}|^6$ in the case of purely rotational noise and $S(|\boldsymbol{q}|) \sim s_{10} |\boldsymbol{q}|^{10}$ in the presence of rototranslational noise. The coefficients $s_6$ and $s_{10}$ can be computed as in the viscous case, to give

$$s_6 = \left( \frac{3}{2} \right)^2 \frac{\rho_0^2 \alpha_6^2 \, \Xi^{(\varphi)}}{\varsigma^2 (\mathcal{D}_2 + \mathcal{D}_6)^3} \,, \tag{41a}$$

$$s_{10} = \left( \frac{3}{4} \right)^2 \frac{\rho_0^2 \alpha_6^2 \, \Xi^{(\varphi)}}{\varsigma^4 (\mathcal{D}_2 + \mathcal{D}_6)^3} \,. \tag{41b}$$

## Numerical methods

### The Voronoi model

In the self-propelled Voronoi model (**Bi et al., 2016**) a confluent cell layer is approximated as a Voronoi tessellation of the plane. Each cell is characterized by the position $\boldsymbol{r}_c$ of its center, with $c = 1, 2 \ldots N$, and a velocity $\boldsymbol{v}_c = v_0 (\cos \theta_c \, \boldsymbol{e}_x + \sin \theta_c \, \boldsymbol{e}_y)$, with $v_0$ a constant speed and $\theta_c$ an orientation. We stress that, in general, the center of a Voronoi polygon does not correspond to the polygon's centroid (i.e. center of mass). The dynamics of these variables is governed by the following set of overdamped Langevin equations, expressing the interplay between cells' autonomous motion and the remodeling events that underlie the tissue's collective dynamics. That is:

$$\frac{\mathrm{d}\boldsymbol{r}_c}{\mathrm{d}t} = \boldsymbol{v}_c - \mu \nabla_{\boldsymbol{r}_c} E \,, \tag{42a}$$

$$\frac{\mathrm{d}\theta_c}{\mathrm{d}t} = \eta_c \,, \tag{42b}$$

where $\mu$ is the mobility coefficient and $E = E(\boldsymbol{r}_1, \boldsymbol{r}_2 \ldots \boldsymbol{r}_N)$ is an energy function involving exclusively geometrical quantities, such as the area $A_c$ and the perimeter $P_c$ of each cell: i.e.

$$E = \sum_c \left[ K_A \left( A_c - A_0 \right)^2 + K_P \left( P_c - P_0 \right)^2 \right] \,, \tag{43}$$

with $K_A$, $K_P$, $A_0$, and $P_0$ constants. The first term in **Equation 43** embodies a combination of cells' volumetric incompressibility and monolayer resistance to thickness fluctuations. The second term results from the cytoskeletal contractility (quadratic in $P_c$) and the effective interfacial tension caused by the cell–cell adhesion and the cortical tension (both linear in $P_c$) (**Farhadifar et al., 2007**). The constants $A_0$ and $P_0$ represent, respectively, the preferred area and perimeter of each cell. The quantity $\eta_c$, on the other hand, is a random number with zero mean and correlation function

$$\langle \eta_c(t) \eta_{c'}(t') \rangle = 2\mathcal{D}_{\mathrm{r}} \delta_{cc'} \delta(t - t') \,, \tag{44}$$

with $\mathcal{D}_r$ a rotational diffusion coefficient. To make progress, we next introduce the following dimensionless numbers: the shape index $p_0 = P_0/\sqrt{A_0}$, which accounts for the spontaneous degree of acircularity of individual cells (**Bi et al., 2016**), and the Péclet number $\mathrm{Pe} = v_0/(\mathcal{D}_r\sqrt{A_0})$, which quantifies the persistence of directed cellular motion in front of their diffusivity.

To obtain the plots in **Figure 3**, we numerically integrate **Equation 42a** in a domain of size $L_g$ with periodic boundary conditions. At $t = 0$, the centroids $\boldsymbol{r}_c$ are placed in a slightly perturbed hexagonal grid with a random initial velocity. After reaching the non-equilibrium steady state, we perform statistical averages of relevant observables.

In our numerical simulations, we set $p_0 = 3.85$, $\mu K_A A_0/\mathcal{D}_r = 1$, $\mu K_P/\mathcal{D}_r = 1$, and $\mathcal{D}_r\Delta t = 5 \times 10^{-3}$, where $\Delta t$ is the time-step used for the integration, and the average density of particles $NA_0/L_g^2 = 1$. We vary the Péclet number in the range $0.1 \leq \mathrm{Pe} \leq 2.0$. The results presented in Results are robust to the variation of the system size, as no qualitative difference was observed upon varying the domain size in the range $30 \leq L_g \leq 200$ at constant density. The density structure factor (light green circles) in **Figure 3a** was obtained, in particular, with $\mathrm{Pe} = 1.5$.

## The MPF model

The MPF model is a continuous model where each cell is described by a concentration field $\varphi_c = \varphi_c(\boldsymbol{r})$ with $c = 1, 2 \ldots N$ and $N$ the total number of cells. This model has been used to study the dynamics of confluent cell monolayers (**Loewe et al., 2020**) and the mechanics of cell extrusion (**Monfared et al., 2021**). Equilibrium configurations are obtained upon relaxing the free energy $\mathcal{F} = \int \mathrm{d}A f$, where the free energy density $f$ is given by

$$f = \frac{\alpha}{4}\sum_c \varphi_c^2(\varphi_c - \varphi_0)^2 + \frac{k_\varphi}{2}\sum_c (\nabla\varphi_c)^2 + \epsilon\sum_{c<c'} \varphi_c^2\varphi_{c'}^2 + \sum_c \lambda\left(1 - \frac{1}{\pi\phi_0^2 R_\varphi^2}\int \mathrm{d}A\,\varphi_c^2\right)^2 . \quad (45)$$

Here, $\alpha$ and $k_\phi$ are material parameters which can be used to tune the surface tension $\gamma = (8k_\varphi\alpha/9)^{1/2}$ and the interfacial thickness $\xi = (2k_\varphi/\alpha)^{1/2}$ of isolated cells and thermodynamically favor spherical cell shapes. The constant $\epsilon$ captures the repulsion between cells. The concentration field is large (i.e. $\varphi_c \simeq \phi_0$) inside the cells and zero outside. The contribution proportional to $\lambda$ in the free energy enforces cell incompressibility whose nominal radius is given by $R_\varphi$. The relaxational dynamics of the field $\varphi_c$ is governed by the Allen–Cahn equation

$$\partial_t\varphi_c + \boldsymbol{v}_c \cdot \nabla\varphi_c = -M\frac{\delta\mathcal{F}}{\delta\varphi_c} , \quad (46)$$

where $\boldsymbol{v}_c$ has the same meaning as in the SPV model described in the previous section and its dynamics is also governed by **Equation 42b**. The constant $M$ in **Equation 46** is the mobility measuring the relevance of thermodynamic relaxation with respect to non-equilibrium cell migration. The dimensionless parameters of the model are the Péclet number $\mathrm{Pe} = v_0/(2\mathcal{D}_r R_\varphi)$ and the cell deformability $d = \epsilon/\alpha$.

The system of partial differential equations, **Equation 46**, is solved with a finite-difference approach through a predictor–corrector finite-difference Euler scheme implementing second-order stencil for space derivatives (**Carenza et al., 2019**). The C-code implemented for numerical integration is parallelized by means of Message Passage Interface (MPI). We consider systems of $N = 361$ cells in a square domain of $L_g = 380$ grid points. Model parameters in simulation units are as follows: $R_\phi = 11$, $\varphi_0 = 2.0$, $M\alpha = 0.006$, $Mk_\varphi = 0.006$, $M\epsilon = 0.01$, $M\lambda = 600$, $M\gamma = 0.008$, $D_r\Delta t = 10^{-4}$, being $\Delta t$ the time-step used to integrate **Equation 46**. We vary the speed of self-propulsion in the range $0.0 \leq v_0 \leq 0.005$. In terms of dimensionless parameters this corresponds to having $d = 1.66$ and $\mathrm{Pe}$ ranging between 0 and 2.30. The timescale of cell motility with respect to the timescale of elastic relaxation driven by surface tension $v_0/(M\gamma)$ ranges between 0 and 0.625. Moreover, the nominal packing fraction is $N(\pi R_\varphi^2)/L_g^2 = 0.95$, while the ratio between the interface thickness and the nominal radius $\xi/R_\varphi = 0.12$. The density structure factor (dark green triangles) in **Figure 3a** was obtained with $\mathrm{Pe} = 1.38$.

## Numerical method for integration of the hydrodynamic equations

**Equation 2a** has been integrated by means of a hybrid lattice Boltzmann (LB) method, in which **Equation (2b)** is solved through a predictor–corrector LB algorithm and the remaining equations via a

predictor–corrector finite-difference Euler approach, with a first-order upwind scheme and second-order accurate stencils for the computation of spacial derivatives (*Carenza et al., 2019*). The code has been parallelized by means of MPI, by dividing the computational domain in slices and by implementing the ghost-cell method to compute derivatives on the boundary of the computational subdomains. Runs have been performed using 64 CPUs in two-dimensional geometries, on a computational box of size $256^2$ and $512^2$, for at least $1.5 \times 10^7$ LB iterations (corresponding to ~21 and ~84 days of CPU-time, respectively, for the smaller and larger computational boxes). Periodic boundary conditions have been imposed. The director fields (for both $p = 2$ and $p = 6$) have been randomly initialized. The initial density field is assumed to be uniform with $\rho = 2.0$ everywhere. The model parameters in simulations units are as follows: $\eta = \zeta = 1.66$, $\lambda_2 = \lambda_6 = 1.1$, $\nu_2 = \nu_6 = 0.0$, $\Gamma_2 = 0.4$, $A_2 = -B_2 = -0.04$, $L_2 = 0.04$, $\Gamma_6 = 0.4$, $A_6 = -B_6 = -0.004$, $L_6 = 0.004$, $\kappa_{2,6} = \xi_{2,6} = -0.004$. Nematic activity $\alpha_2$ has been varied in the range $-0.02 \leq \alpha_2 \leq -0.0005$ and hexatic activity $\alpha_6$ in the range $-0.050 \leq \alpha_6 \leq 0.050$. We set the active parameters $\beta_2$ and $\beta_6 = 0$. The density structure factor (continuous black line) in *Figure 3a* was obtained with $\alpha_2 = -2 \times 10^{-3}$ and $\alpha_6 = 2 \times 10^{-2}$.

The coherence length of the nematic and hexatic liquid crystal can be expressed as the $(L_p/A_p)^{1/2} = \Delta x_{\text{LB}}$ for both $p = 2, 6$, where $\Delta x_{\text{LB}}$ is the grid spacing of the LB algorithm. The active length scale as defined in the main text is given for the active nematics as $\ell_2$ and ranges between $10\Delta x_{\text{LB}}$ for $\alpha_2 = -0.0005$ and $1.5\Delta x_{\text{LB}}$ for $\alpha_2 = -0.02$. Conversely, for hexatics $\ell_6$ and ranges up to $3.5\Delta x_{\text{LB}}$ for $|\alpha_6| = 0.05$. To compare the results of the hydrodynamics simulations with the discrete models in *Figure 3a*, we choose $2\Delta x_{\text{LB}} = \sqrt{A_0}$ and $2\Delta x_{\text{LB}} = R_\varphi \Delta x_{\text{MP}}$, with $\Delta x_{\text{MP}}$ the grid spacing used to integrate *Equation 46*.

## Comparison with passive liquid crystals with coupled order parameters

In this section, we show how multiscale *hexanematic* order differs from previously reported examples of liquid crystal order with coupled order parameters (*Bruinsma and Aeppli, 1982*; *Selinger and Nelson, 1989*; *Selinger, 1991*). To quantify the interplay between nematic and hexatic order, here we focus on the function $C_{26}(\boldsymbol{r})$ given in *Equation 9*, reflecting the amount of cross-correlation in their fluctuations. Here, $\psi_2 = e^{2i\vartheta}$ and $\psi_6 = e^{6i\varphi}$, while the fluctuating fields $\vartheta$ and $\varphi$ represent again the local nematic and hexatic orientations, respectively. Averaging $\psi_2$ and $\psi_6$ over the scale of a volume element, yields the order complex parameters $\Psi_2 = \langle e^{2i\vartheta} \rangle = |\Psi_2|e^{2i\theta}$ and $\Psi_6 = \langle e^{6i\varphi} \rangle = |\Psi_6|e^{6i\phi}$, with $\theta$ and $\phi$ the average orientations. To make progress, we assume that, at the scale of a volume element, both microscopic orientations $\vartheta$ and $\varphi$ are Gaussianly distributed about their mean values, so that, in general

$$\Psi_p = \langle \psi_p \rangle \approx e^{-\frac{1}{2}\,\text{var}[\text{Arg}(\psi_p)] + i\langle\text{Arg}(\psi_p)\rangle}\,, \tag{47}$$

from which

$$|\Psi_p| \approx e^{-\frac{1}{2}\,\text{var}[\text{Arg}(\psi_p)]}\,, \qquad \text{Arg}(\Psi_p) = \langle\text{Arg}(\psi_p)\rangle\,. \tag{48}$$

This approximation holds when the relative fluctuation of the *p*-atic phase $\text{Arg}(\psi_p)$ is sufficiently small, so that

$$|\Psi_p| \approx 1 - \frac{1}{2}\left\langle \left[\text{Arg}(\psi_p) - \text{Arg}(\Psi_p)\right]^2 \right\rangle \approx \left\langle \cos\left[\text{Arg}(\psi_p) - \text{Arg}(\Psi_p)\right]\right\rangle\,, \tag{49}$$

consistent with the standard definition of *p*-atic order parameter. Thus, in particular, $\theta = \langle\vartheta\rangle$ and $|\Psi_2| = \langle\cos 2(\vartheta - \theta)\rangle$, whereas $\phi = \langle\varphi\rangle$ and $|\Psi_6| = \langle\cos 6(\varphi - \phi)\rangle$. This allows to write $C_{26}(\boldsymbol{r})$, as given by *Equation (9)*, in the form

$$C_{26}(\boldsymbol{r}) = \frac{\Psi_2(\boldsymbol{r})\Psi_6^*(\boldsymbol{0}) + \Psi_2^*(\boldsymbol{r})\Psi_6(\boldsymbol{0})}{2}\,e^{12\left[\langle\vartheta(\boldsymbol{r})\varphi(\boldsymbol{0})\rangle - \langle\vartheta(\boldsymbol{r})\rangle\langle\varphi(\boldsymbol{0})\rangle\right]}\,. \tag{50}$$

At equilibrium, both nematic and hexatic order can be approximated as uniform, so that

$$\frac{\Psi_2(\boldsymbol{r})\Psi_6^*(\boldsymbol{0}) + \Psi_2^*(\boldsymbol{r})\Psi_6(\boldsymbol{0})}{2} = |\Psi_2||\Psi_6|\cos(2\theta - 6\phi) \approx \text{const}\,, \tag{51}$$

and the problem reduces to calculating the connected correlation function

$$C_{\vartheta\varphi}(\boldsymbol{r}) = \langle\vartheta(\boldsymbol{r})\varphi(\boldsymbol{0})\rangle - \langle\vartheta(\boldsymbol{r})\rangle\langle\varphi(\boldsymbol{0})\rangle \ . \tag{52}$$

Notice that *Equation (51)* is not strictly valid for a quasi long-ranged ordered liquid crystal, where also $\theta$ and $\phi$ are expected to vary in space. These spatial variations, however, occur on length scales comparable with the system size and, as long as this is much larger than any of the intrinsic length scales entailed in *Equation 2a*, are negligible for the purpose of this calculation. To compute $C_{\vartheta\varphi}(\boldsymbol{r})$, one can take the passive limit of *Equation 2c* and linearize the resulting equations about the lowest free energy configuration. This, in turn, is determined by the sign of the constant $\chi_{2,6}$ in *Equation 6b*. For $\chi_{2,6} < 0$, the hexatic and nematic directors are energetically favored to be parallel, so that $\vartheta \approx \varphi$. Conversely, when $\chi_{2,6} > 0$, the hexatic and nematic directors are preferentially tilted by $\pi/6$, hence $\vartheta = \varphi \pm \pi/6$. For presentational clarity, here we focus on the former case and, at the end of this section, we show how the same behavior holds for positive $\chi_{2,6}$ values. Thus, assuming $\chi_{2,6} < 0$ and expanding *Equation 2c* about $\vartheta \approx \varphi$, gives

$$\partial_t\vartheta = \mathcal{D}_2\nabla^2\vartheta - \frac{9}{4}|\chi_2|\left(\vartheta - \varphi\right) + \xi^{(\vartheta)} \ , \tag{53a}$$

$$\partial_t\varphi = \mathcal{D}_6\nabla^2\varphi - \frac{1}{4}|\chi_6|\left(\varphi - \vartheta\right) + \xi^{(\varphi)} \tag{53b}$$

where, as in the previous sections, we have set $\mathcal{D}_p = \Gamma_p L_p$ and $\chi_p = \Gamma_p \chi_{2,6}$ and introduced the Gaussian noise fields $\xi^{(\vartheta)}$ and $\xi^{(\vartheta)}$, having vanishing mean and finite variance. Unlike the active case, however, at equilibrium the latter is related to the environmental temperature by the fluctuation–dissipation theorem. This implies

$$\left\langle\xi^{(\alpha)}(\boldsymbol{r},t)\xi^{(\beta)}(\boldsymbol{r}',t')\right\rangle = 2k_BT\left(\frac{\delta_{\alpha\vartheta}\delta_{\beta\vartheta}}{\gamma_2} + \frac{\delta_{\alpha\varphi}\delta_{\beta\varphi}}{\gamma_6}\right)\delta(\boldsymbol{r}-\boldsymbol{r}')\delta(t-t') \ , \tag{54}$$

where $\gamma_p = K_p/\mathcal{D}_p$, with $K_p$ the orientational stiffness defined in *Equation 15*, is the rotational viscosity of the associated $p$-atic phase. *Equation 53a* can now be decoupled and used to compute the correlation function $C_{\vartheta\varphi}(\boldsymbol{r})$. For simplicity, here we set $\mathcal{D}_2 = \mathcal{D}_6 = \mathcal{D}$, $\gamma_2 = \gamma_6 = \gamma$, and $9\chi_2 = \chi_6 = 2\chi$. With this choice, taking

$$\varphi_+ = \frac{1}{2}\left(\varphi + \vartheta\right) \ , \tag{55a}$$

$$\varphi_- = \frac{1}{2}\left(\varphi - \vartheta\right) \ , \tag{55b}$$

gives, after simple algebraic manipulations

$$\partial_t\varphi_+ = \mathcal{D}\nabla^2\varphi_+ + \xi_+ \ , \tag{56a}$$

$$\partial_t\varphi_- = \mathcal{D}\nabla^2\varphi_- - |\chi|\varphi_- + \xi_- \ , \tag{56b}$$

where $\xi_+ = (\xi^{(\varphi)} + \xi^{(\vartheta)})/2$ and $\xi_- = (\xi^{(\varphi)} - \xi^{(\vartheta)})/2$. Moreover, using *Equation (54)*, one finds

$$\left\langle\xi_n(\boldsymbol{r},t)\xi_m(\boldsymbol{r}',t')\right\rangle = \frac{2k_BT}{\gamma}\delta_{nm}\delta(\boldsymbol{r}-\boldsymbol{r}')\delta(t-t') \ , \tag{57}$$

where $\{n,m\} = \{+,-\}$. *Equation 56a* can now be solved in Fourier space and real time to give

$$\hat{\varphi}_n(\boldsymbol{q},t) = e^{S_n(\boldsymbol{q},t)}\left[\hat{\varphi}_n(\boldsymbol{q},0) + \int_0^t \mathrm{d}t' \, e^{-S_n(\boldsymbol{q},t')}\hat{\xi}_n(\boldsymbol{q},t')\right] \ , \tag{58}$$

where the hat indicates Fourier transformation and

$$S_n(\boldsymbol{q},t) = -\mathcal{D}t\left(|\boldsymbol{q}|^2 + m_n^2\right) \ , \tag{59}$$

where $m_+ = 0$ and $m_-^2 = \ell_\chi^{-2} = \mathcal{D}/|\chi|$. The calculation of the cross-correlation function $C_{\vartheta\varphi}(\boldsymbol{r})$ is now reduced to calculating the autocorrelation functions of the fields $\varphi_+$ and $\varphi_-$. Specifically

$$C_{\vartheta\varphi}(\boldsymbol{r}) = C_{++}(\boldsymbol{r}) - C_{--}(\boldsymbol{r}) \,, \tag{60}$$

where

$$C_{nm}(\boldsymbol{r}) = \langle \varphi_n(\boldsymbol{r})\varphi_m(\boldsymbol{0}) \rangle - \langle \varphi_n(\boldsymbol{r}) \rangle \langle \varphi_m(\boldsymbol{0}) \rangle \,, \tag{61}$$

and we have made use of **Equation (54)** to demonstrate that $C_{+-}(\boldsymbol{r}) = C_{-+}(\boldsymbol{r}) = 0$. The non-vanishing correlation functions, on the other hand, can be expressed as

$$C_{nn}(\boldsymbol{r}) = \lim_{t\to\infty} \int_{0<|\boldsymbol{q}|<\Lambda} \frac{\mathrm{d}^2 q}{(2\pi)^2} \, e^{i\boldsymbol{q}\cdot\boldsymbol{r}} \langle |\hat{\varphi}_n(\boldsymbol{q},t)|^2 \rangle, \tag{62}$$

where $\Lambda = 2\pi/a$ is a short-distance cut-off and $\langle |\hat{\varphi}_n(\boldsymbol{q},t)|^2 \rangle$ is the finite-time orientational structure factor defined from the relation

$$\langle \hat{\varphi}_n(\boldsymbol{q},t)\hat{\varphi}_n(\boldsymbol{q},t') \rangle = (2\pi)^2 \langle |\hat{\varphi}_n(\boldsymbol{q},t)|^2 \delta(\boldsymbol{q}+\boldsymbol{q}')\delta(t-t') \,. \tag{63}$$

After standard algebraic manipulations one finds

$$\langle |\hat{\varphi}_n(\boldsymbol{q})|^2 \rangle = \lim_{t\to\infty} \langle |\hat{\varphi}_n(\boldsymbol{q},t)|^2 \rangle = \frac{k_B T}{K} \frac{1}{|\boldsymbol{q}|^2 + m_n^2} \,. \tag{64}$$

from which **Equation (62)** can be calculated in the form

$$C_{nn}(\boldsymbol{r}) = \frac{k_B T}{K} \int_{0<|\boldsymbol{q}|<\Lambda} \frac{\mathrm{d}^2 q}{(2\pi)^2} \frac{e^{i\boldsymbol{q}\cdot\boldsymbol{r}}}{|\boldsymbol{q}|^2 + m_n^2} \,. \tag{65}$$

Evidently, **Equation (65)** is equivalent to that obtained in a purely static setting from the Hamiltonian

$$\mathcal{H} = \frac{1}{2} \int \mathrm{d}^2 r \left[ K|\nabla\varphi_+|^2 + K|\nabla\varphi_-|^2 + m_-^2 \varphi_-^2 \right] \,, \tag{66}$$

of the non-interacting scalar fields $\varphi_+$ and $\varphi_-$. Now, in the case of the 'massive' field $\varphi_-$, the Fourier integral in **Equation (65)** converges to

$$C_{--}(\boldsymbol{r}) = \frac{k_B T}{2\pi K} K_0\left( \frac{|\boldsymbol{r}|}{\ell_\chi} \right) \,, \tag{67}$$

in the range $|\boldsymbol{r}| \gg a$. Here, $K_0$ is a modified Bessel function of the second kind, whose asymptotic expansion at short and long distances is given by

$$K_0(z) \approx \begin{cases} -\gamma_{\mathrm{EM}} - \log \dfrac{z}{2} & 0 < z \ll 1 \,, \\[2mm] \sqrt{\dfrac{\pi}{2z}} \, e^{-z} & z \gg 1 \end{cases} \,, \tag{68}$$

with $\gamma_{\mathrm{EM}}$ the Euler–Mascheroni constant. In the case of the 'massless' field $\varphi_+$, on the other hand, the Fourier integral diverges in the infrared, but the correlation function $C_{++}(\boldsymbol{r})$ can still be computed as the Laplacian Green function on an infinite domain punctured by a hole of radius $a$ at the origin. Thus

$$C_{++}(\boldsymbol{r}) = -\frac{k_B T}{2\pi K} \log \frac{|\boldsymbol{r}|}{a} \,. \tag{69}$$

Combining this with **Equations (67) and (69)** yields the following expression for the correlation function

$$C_{\vartheta\varphi}(\boldsymbol{r}) = -\frac{k_B T}{2\pi K} \left[ \log \frac{|\boldsymbol{r}|}{a} + K_0\left( \frac{|\boldsymbol{r}|}{\ell_\chi} \right) \right] \,, \tag{70}$$

where $|\boldsymbol{r}| \gg a$. Finally, using **Equation (50)** and the asymptotic expansions of **Equation (68)** gives the following expression for the cross-correlation function

$$C_{26}(\boldsymbol{r}) \sim \begin{cases} \text{const.} & |\boldsymbol{r}| \ll \ell_\chi \\[2mm] \left(\dfrac{|\boldsymbol{r}|}{a}\right)^{-\eta_{26}} & |\boldsymbol{r}| \gg \ell_\chi \end{cases}, \tag{71}$$

where $\eta_{26}$ is an instance of the generic non-universal exponent

$$\eta_{pp'} = \frac{pp' k_B T}{2\pi K}, \tag{72}$$

in the specific case $p = 2$ and $p' = 6$. Lastly, when $\chi_{2,6} > 0$, the same procedure can be carried out by expanding *Equation (2c)* about $\vartheta = \varphi \pm \pi/6$ and taking $\varphi_+ = (\varphi + \vartheta)/2$ and $\varphi_- = (\varphi - \vartheta \pm \pi/6)/2$, from which one finds again *Equation 72*.

## Acknowledgements

We are indebted with Massimo Pica Ciamarra and David Nelson for insightful discussions. This work is supported by the ERC-CoG grant HexaTissue (LNC and LG) and by Netherlands Organization for Scientific Research (NWO/OCW) as part of the research program 'The active matter physics of collective metastasis' with project number Science-XL 2019.022 (J-M A-C). Part of this work was carried out on the Dutch national e-infrastructure with the support of SURF through the NWO Grant 2021.028 for computational time.

## Additional information

### Funding

| Funder | Grant reference number | Author |
|---|---|---|
| European Research Council | | Livio Nicola Carenza Luca Giomi |
| Nederlandse Organisatie voor Wetenschappelijk Onderzoek | | Josep-Maria Armengol-Collado |

The funders had no role in study design, data collection, and interpretation, or the decision to submit the work for publication.

### Author contributions

Josep-Maria Armengol-Collado, Conceptualization, Data curation, Formal analysis, Visualization, Writing - original draft; Livio Nicola Carenza, Conceptualization, Data curation, Formal analysis, Supervision, Visualization, Writing - original draft; Luca Giomi, Conceptualization, Formal analysis, Supervision, Methodology, Writing - original draft, Project administration

### Author ORCIDs

Josep-Maria Armengol-Collado http://orcid.org/0000-0003-0740-3040
Livio Nicola Carenza http://orcid.org/0000-0001-5996-331X
Luca Giomi http://orcid.org/0000-0001-7740-5960

### Decision letter and Author response

Decision letter https://doi.org/10.7554/eLife.86400.sa1
Author response https://doi.org/10.7554/eLife.86400.sa2

### Data availability

Figures 2–4 contain the numerical data used to generate the figures.

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
