## [Editor Report]

This important work presents a hydrodynamic description of confluent epithelial monolayers that captures different forms of orientational order in a scale dependent fashion and couples this order with flows driven by active stresses. Solid evidence for the validity of this approach is provided by detailed numerical simulations of different model tissues. This work should be of interest to a broad range of biophysicists interested in tissue mechanics and active matter.

---

## [Decision Letter]

**Decision letter after peer review:**

Thank you for submitting your article "Hydrodynamics and multiscale order in confluent epithelia" for consideration by *eLife*. Your article has been reviewed by 2 peer reviewers, and the evaluation has been overseen by a Reviewing Editor and Aleksandra Walczak as the Senior Editor. We regret the long delay in furnishing this decision letter.

The reviewers have discussed their reviews with one another, and the Reviewing Editor has drafted this to help you prepare a revised submission. As you can see from the reviews, the referees are indeed supportive of your work, but in their very detailed reports, they request clarification on a number of points.

Essential revisions:

1) We find the tensorial notation used by the authors complicated, and would prefer to see tensor terms written with indices for clarity, but I acknowledge that this may be a matter of personal preference.

2) Page 6 – the manuscript is referring to panel 2d which we think is not in the figure.

3) We are not sure about the statement that the p-dependent structure makes the multi-scale nature of the system « enormously more dramatic »; as in many physical problems, several characteristic length scales are involved; It is unclear what the authors mean with this strong statement.

4) We are unconvinced by the argument that the short wavelength scaling of density fluctuations is necessarily a fingerprint of small-scale hexatic order. The claim is all the more problematic if Eq. 8 is a small q expansion, in which case it is unclear if we should trust the large q-scaling. It is also entirely unclear how and why other non-hydrodynamic modes will not contribute to the scaling at large *q*. Eventually, at a single-cell level, all orientational modes should become important, not just the nematic and the hexatic ones. One way to point out the distinct imprint of hexatic order is to not simply plot *S(q)* for fixed *qx=qy* or angle-averaged, but rather to consider the full 2D plot in q-space. The nematic contribution has a characteristic singularity ~ *q_x_ q_y_/q^4^* at small q. The hexatic contribution should also give rise to a particular anisotropic structure in *S(q)* that could serve as a diagnostic and strengthen the point made in the paper.

A useful reference and point of comparison in the equilibrium context is – Aeppli, G., and R. Bruinsma. "Hexatic order and liquid density fluctuations." Physical Review Letters 53.22 (1984): 2133.

5) The paper can also benefit by making a stronger comparison with existing literature on models for equilibrium liquid crystals with coupled order parameters that offer a null model. Some references are mentioned below.

Dynamics, thermodynamics, and optical scattering of tilted hexatics (coupled hexatic+polar tilt order, but much of the physics is very similar)

Dierker, S. B., and R. Pindak. "Dynamics of thin tilted hexatic liquid crystal films." Physical review letters 59.9 (1987): 1002.

Sprunt, S., and J. D. Litster. "Light-scattering study of bond orientational order in a tilted hexatic liquid-crystal film." Physical review letters 59.23 (1987): 2682.

Selinger, Jonathan V. "Dynamics of tilted hexatic phases in liquid-crystal films." Journal de Physique II 1.11 (1991): 1363-1373

Selinger, Jonathan V., and David R. Nelson. "Theory of transitions among tilted hexatic phases in liquid crystals." Physical Review A 39.6 (1989): 3135.

6) Coupled hexatic-nematic models appear in certain liquid crystals (see e.g., Bruinsma, R., and G. Aeppli. "Hexatic order and herring-bone packing in liquid crystals." Physical Review Letters 48.23 (1982): 1625.) These models can have unusual emergent Potts phases, where both order parameters are disordered with finite correlation lengths, but the relative angles between hexatic-nematic order parameters remain ordered – see recent work.

Drouin-Touchette, Victor, et al. "Emergent potts order in a coupled hexatic-nematic xy model." Physical Review X 12.1 (2022): 011043.

It may be interesting to ask if similar phenomena can occur in the active variant studied in this paper and if it may be potentially relevant for tissues.

7) It is extremely unclear how the hexatic order parameter is microscopically defined; there seems to be no description given. Is the hexatic order parameter constructed from cell shape or from the distribution of `bonds' to neighboring cells? Are the two definitions equivalent when confluent? Presumably, the shape and bond angle hexatic order parameters can differ in a dense system of deformable particles that are not close-packed/confluent. The authors should comment on what the origin of the hexatic ordering is and how one may compute such an order parameter in reality.

On a related point, we fail to see why order parameter correlations are not measured and plotted to demonstrate in a straightforward way the claim of hexatic and nematic order being present on different length scales. Such calculations are presented elsewhere in recent preprints by some of the authors when analyzing experimental data though.

8) Although emphasizing multiscale aspects, the model crucially neglects any mechanism of feedback. Active couplings other than the active stress, e.g., nonreciprocal cross terms aligning the two order parameters, are neglected, though they are likely to be more dominant in a gradient expansion compared to the hexatic contribution to the active stress.

Furthermore, while substrate friction is included, traction forces (polar active forces) that are commonly exerted by cells are also neglected, which could presumably also dominate over hexatic degrees of freedom, even when randomized.

9) Equation 3-5: It would be helpful if the notation could be fixed and either σ^a^ or Pi is used. The relationship between the two is quite confusing at the moment.

Also, sign conventions should be explained: is α_2_>0 contractile, and if so then does the relation with f have an extra minus sign? Also, Eq. 5 suggests that α_6_ should have the same sign as α_2_, yet in the simulations opposite signs are chosen. What is its meaning and the relation between isotropic/hexatic and nematic extensile or contractile activity?

It would also help to point out how the ratio of l2 and l6 (Eq. 7) differ by a ratio of the cell size to the conventional active nematic length (l2).

The sign of ξ2,6 affects the alignment between Q2 and Q6, how does its sign affect the density and orientational correlations?

---

## [Author Response]

Essential revisions:1) We find the tensorial notation used by the authors complicated, and would prefer to see tensor terms written with indices for clarity, but I acknowledge that this may be a matter of personal preference.

Unfortunately this complication is unavoidable, as it is inherent to generic rank−*p* tensors, with *p* an arbitrary integer. Because of this arbitrariness, expressing these tensors in index notation would require using, e.g., Qi1i2⋯ip and a similar punctuated notation throughout the manuscript, making the equations even less accessible. Eq. (2c), for instance, would appear as DQi1i2…ipDt=ΓpHi1i2…ip+p[[Qi1i2⋯jωjip]]+λ¯pQi1i2⋯ip+λp[[∂i1∂i2⋯∂ip−2uip−1ip]]+vp{[[ui1i2ui3i4⋯uip−1ip]]peven[[∂i1ui2i3ui4i5⋯uip−1ip]]podd .While this may facilitate further manipulations, we are afraid it will be at the expenses of the casual reader, who can now readily recognize the physical meaning of the various terms in Eq. (2c), as well as the order of the differential operators acting upon them, without having to decipher the index notation.

2) Page 6 – the manuscript is referring to panel 2d which we think is not in the figure.

We are sorry for the typo. We have now changed 2d to 2b.

3) We are not sure about the statement that the p-dependent structure makes the multi-scale nature of the system « enormously more dramatic »; as in many physical problems, several characteristic length scales are involved; It is unclear what the authors mean with this strong statement.

The statement applies to passive liquid crystals. We are in fact not aware of any liquid crystal at equilibrium where multiple *p*−atic orders characterize different length scales. This includes liquid crystals with coupled order parameters, such as the herring-bone packings investigated by Bruinsma, Selinger and others, where both nematic and hexatic order coexist at the same length scale. Epithelial layers differ from this picture in two essential aspects. (1) Being confluent assemblies of polydisperse *irregular* polygons, violate the standard one-to-one correspondence between microscopic shape and macroscopic order (e.g. rod-like shape ↔ nematic order, triangular shape ↔ triatic order etc.), thereby making more complex types of organization possible. (2) The combination between *p*−atic order and activity gives rise to active stresses of the form given in Eq. (3), where multiscale organization is inherent, because the integer *p* determines *both* the rotational symmetry of the ordered phase and the order of the differential operator. These stresses drives currents which, in turn, give rise to features reflecting the symmetry of the corresponding ordered phase and here are described in terms of the static structure factor.

Now, property *1)* is only partially captured by the approach presented here, as this is still based on the orientational order parameter *ψ_p_* = *e^ipϑ^*, with *ϑ* the local orientation of the building blocks. An alternative treatment, used in Armengol-Collado *et al.*, Nat. Phys. (2023) and Eckert *et al.* Nat. Commun. (2023), revolves around the “shape function” *γ_p_* (see also Appendix 5 of the revised manuscript). The latter entails more information than *ψ_p_* and can be used to demonstrate that, aside from the spatial features driven by the active flow, hexatic order is in fact more prominent at short length scales – because of the approximatively hexagonal shape of the cells – while nematic order at large length scales – as a consequence of how forces propagate throughout the cell layer. We have now elaborated on these concepts in the *Introduction* and in the section Multiscale order in epithelia.

4) We are unconvinced by the argument that the short wavelength scaling of density fluctuations is necessarily a fingerprint of small-scale hexatic order. The claim is all the more problematic if Eq. 8 is a small q expansion, in which case it is unclear if we should trust the large q-scaling. It is also entirely unclear how and why other non-hydrodynamic modes will not contribute to the scaling at large q. Eventually, at a single-cell level, all orientational modes should become important, not just the nematic and the hexatic ones. One way to point out the distinct imprint of hexatic order is to not simply plot *S(q)* for fixed *q*_*x*_*=q*_*y*_ or angle-averaged, but rather to consider the full 2D plot in q-space. The nematic contribution has a characteristic singularity ~ *q_x_ q_y_/q^4^* at small *q*. The hexatic contribution should also give rise to a particular anisotropic structure in *S(q)* that could serve as a diagnostic and strengthen the point made in the paper.A useful reference and point of comparison in the equilibrium context is – Aeppli, G., and R. Bruinsma. "Hexatic order and liquid density fluctuations." Physical Review Letters 53.22 (1984): 2133.

This is indeed a delicate point. Before elaborating it further, we would like to stress that looking at the large *q* limit is less unusual in interfacial phenomena and liquid crystals than in other research areas. One of the best known result about the large *q* limit of the structure factor is the celebrated Porod law, which was generalized to two-dimensional nematic liquid crystals by Zapotocky and Goldbart (see arXiv:cond-mat/9812235). The reason why neither Porod’s law nor our calculation is problematic, is because in both cases the structure factor has an *exact* asymptotic scaling form (see Figure 2 in the revised manuscript), which allows one to compute the *q* → ∞ limit exactly, rather than by arbitrarily truncating a power series. Kolmogorov −5*/*3 law is another classic example of an asymptotically exact scaling form emerging at short length scales, in this case from the spectrum of velocity rather than density fluctuations.

In the present case, our interpretation is further corroborated by the fact that the exponent *β*, dictating the short wavelength scaling of the structure factor, is not universal, but crossovers from *β* = 6 – corresponding to the overdamped limit, when friction is the only momentum-dissipation mechanism at play – to *β* = 4, when momentum is dissipated through viscosity. This is shown in Figure 3b (Figure 2b in the previous version of the manuscript), obtained from numerical simulations of two very different discrete models of epithelia. Our simple linear calculations does not allow to capture the full crossover, but yields a precise computation of the upper and lower bounds of this range. We find the latter too specific to be a mere coincidence.

Furthermore, because of the lack of long-ranged order, the dependence of the structure factor on the individual components of the wave vectors – i.e. *q_x_* and *q_y_* – is a well known artefact of linearizing the hydrodynamic equations about a specific orientation (i.e. *ϑ* = *ϕ* = 0). The latter could be avoided by either linearizing about a pair generic orientations (e.g. *ϑ*_0_ and *ϕ*_0_) and then average over these (i.e. circular average) or, more simply, by orienting *q* is such a way to cancel the directional dependence of the structure factor (i.e. *q_x_* = *q_y_*). Both approaches have been successfully experimented in active nematics [see Ramaswamy *et al.*, EPL 62, 196 (2003) for the former and Shankar *et al.* PRE 97, 012707 (2018) for the latter]. Notice also that, again in active nematics, not taking into account the bias resulting from the initial linearization would lead to the absurd conclusion that the amplitude of the density fluctuations, proportional to qx2qy2/q6, vanished along two different and *fixed* directions.

Lastly, as mentioned in our reply to the previous comment, the existence of hexatic order at the smallscale, was also demonstrated in Armengol-Collado *et al.*, Nat. Phys. (2023) and Eckert *et al.* Nat. Commun. (2023), using a large data set comprising both numerical simulations and experiments on two different MDCKs phenotypes. The main goal of the present paper is *not* to demonstrate the existence of multiscale hexanematic order, whose existence has been already demonstrated, but to harness this peculiar example of physical organization of biological matter within a continuum theory.

Having clarified this, we obviously agree with the general concept that, at short length scales, *any* physical behaviour is much more system-dependent than at large length scales. Inertial turbulence is, once again, a good example of this specificity: i.e. considering momentum-dissipation mechanisms other than viscosity leads to exponents other than −5*/*3. To elaborate on this aspect of the problem, we have now extended our analysis of the *q* → ∞ limit of the structure factor to include four different scenarios, obtained upon combining two different momentum dissipation mechanisms (i.e. viscosity and Stokesian friction) with two different types of noise (i.e. rototranslational and purely rotational). In addition, we have considerably expanded the discussion following the calculation of the structure factor.

5) The paper can also benefit by making a stronger comparison with existing literature on models for equilibrium liquid crystals with coupled order parameters that offer a null model. Some references are mentioned below.Dynamics, thermodynamics, and optical scattering of tilted hexatics (coupled hexatic+polar tilt order, but much of the physics is very similar)Dierker, S. B., and R. Pindak. "Dynamics of thin tilted hexatic liquid crystal films." Physical review letters 59.9 (1987): 1002.Sprunt, S., and J. D. Litster. "Light-scattering study of bond orientational order in a tilted hexatic liquid-crystal film." Physical review letters 59.23 (1987): 2682.Selinger, Jonathan V. "Dynamics of tilted hexatic phases in liquid-crystal films." Journal de Physique II 1.11 (1991): 1363-1373Selinger, Jonathan V., and David R. Nelson. "Theory of transitions among tilted hexatic phases in liquid crystals." Physical Review A 39.6 (1989): 3135.

We have included an overview of liquid crystals with coupled order parameters and included these citations. But we respectfully disagree with the statement about the physics being “very similar”. Multiscale liquid crystal order, as we understand it from this and other investigations from our research group, is a genuinely non-equilibrium phenomenon, resulting from the existence of specific order-dependent active stresses. These can be viewed as a classic Maxwell daemon, organizing biological matter beyond the limitations of thermal equilibrium. In the revised version of the manuscript we have largely elaborated on this concept and provided further characterizations of multiscale orientational order using both analytical and numerical calculations (see our reply to comment # 7 for details).

6) Coupled hexatic-nematic models appear in certain liquid crystals (see e.g., Bruinsma, R., and G. Aeppli. "Hexatic order and herring-bone packing in liquid crystals." Physical Review Letters 48.23 (1982): 1625.) These models can have unusual emergent Potts phases, where both order parameters are disordered with finite correlation lengths, but the relative angles between hexatic-nematic order parameters remain ordered – see recent work.Drouin-Touchette, Victor, et al. "Emergent potts order in a coupled hexatic-nematic xy model." Physical Review X 12.1 (2022): 011043.It may be interesting to ask if similar phenomena can occur in the active variant studied in this paper and if it may be potentially relevant for tissues.

We have included a citation to this paper. Given the differences highlighted above, however, we prefer to avoid digressions on concept that lie well beyond the scope of the manuscript.

7) It is extremely unclear how the hexatic order parameter is microscopically defined; there seems to be no description given. Is the hexatic order parameter constructed from cell shape or from the distribution of `bonds' to neighboring cells? Are the two definitions equivalent when confluent? Presumably, the shape and bond angle hexatic order parameters can differ in a dense system of deformable particles that are not close-packed/confluent. The authors should comment on what the origin of the hexatic ordering is and how one may compute such an order parameter in reality.On a related point, we fail to see why order parameter correlations are not measured and plotted to demonstrate in a straightforward way the claim of hexatic and nematic order being present on different length scales. Such calculations are presented elsewhere in recent preprints by some of the authors when analyzing experimental data though.

We are sorry for the extreme lack of clarity. Some details about our computation of the cell orientation were given, in the original version of the manuscript, in Appendix 5 (Appendix 1 in the revised version of the manuscript), but was evidently too concise to convey the message with the necessary clarity. We have now reviewed, extended and illustrated this appendix to compensate for this lack. In summary, the *p*−atic order parameter is defined in the traditional way, via the ensemble average of the complex function *ψ_p_* = *e^ipϑ^*, with *ϑ* the local orientation of the *p*−atic building blocks. If these are rods or regular polygons, *ϑ* is simply the orientation of the rods or of any of the vertices of the polygons. The *p*−fold orientation of irregular polygons, on the other hand, can be found via the “shape function” *γ_p_* introduced in Armengol-Collado *et al.*, Nat. Phys. (2023). For a *V* −sided polygon, this is given by γp=∑v=1V|r|peipϕv.∑v=1V|r|pwhere ***r**_v_* is marks the position of the *v*−th vertex of the polygon with respect to its center and ϕ*_v_* = Arg(***r**_v_*). The magnitude *γ_p_* of this function quantifies the resemblance between an irregular polygon and a *p*−sided *regular* polygon of the same size, while its phase *ϑ* = Arg(*γ_p_*)*/p* determines the polygon’s orientation.

The reason why we decided not to present the orientational correlation function Cp(r)=⟨ψp∗(r)ψp(0)⟩ is because this function does not convey any information about the multiscale structure of the system, which is instead the main focus of this work. By construction *C_p_*(|***r***| = *a_p_*) = 1 at the microscopic scale *a_p_* and then monotonically decays. The only signal of multiscale order in such a dataset relies on the fact that the notion “microscopic scale” – corresponding to the ultraviolet cut-off the hydrodynamic theory – is order-dependent (i.e. *a*_2_ ≠ *a*_6_) and this could in principle leave some signature on the orientational correlation function at short distances. Unfortunately, being the ultraviolet cut-off a concept of equilibrium physics and not specific of active matter, the same difference is found in principle in any liquid crystal with coupled order parameters. Thus the difference between *mono* and *multi*scale orientational order, when looked through the lens of the orientational correlation function *C_p_*(***r***), simply amounts to a different prominence of the same feature at distances where the signal-to-noise ratio is notoriously small. A much better signal can instead be obtained by coarse-graining the shape function *γ_p_*, to obtain the scale-dependent “shape parameter” Γp(ℓ)=⟨γp⟩ℓ, where h⟨⋯⟩ℓ is an ensemble average over a domain of size *`*. This quantity was also introduced in Armengol-Collado *et al.*, Nat. Phys. (2023) and computed for a large data set comprising the two discrete models used here, as well as experiments on MDCKs cell monolayers on glass. Various other measurements of Γ*_p_*, from epithelial cells having different density, phenotype and plated on substrate of different stiffness, are reported in Eckert *et al.* Nat. Commun. (2023). For this reason, we find unnecessary to repeat the same measurements here on a smaller data set obtained only from numerical simulations.

There is however a different type of correlation function that allows to discriminate between mono and multiscale order. This is the cross-correlation function

C26(r)=⟨ψ2(r)ψ6∗(0)+ψ2∗(r)ψ6(0)⟩2

In the revised version of the manuscript we have computed this quantity analytically, in the case of passive liquid crystals with coupled hexatic-nematic order parameter, and numerically, in both the passive and active case. The difference between these two scenarios is striking and provides one more signature of multiscale orientational order. In a nutshell, in the passive case, the coupling between hexatic and nematic order, expressed by Eq. (6b) in the manuscript, introduces an additional length scale ℓ*_x_*
=D/|x|, with *D* the rotational diffusion coefficient (here assumed for simplicity to be equal in both phases) and *X* a constant expressing the rate at which the hexatic and nematic orientation align with each other. At distances much smaller than ℓ*_x_* fluctuations dominate and hexatic and nematic order are uncorrelated. By contrast, at distances much larger than ℓ*_x_* the local hexatic and nematic orientations are “locked” to each other and the correlation function *C*_26_(***r***) displays the standard power-law decay characterizing two-dimensional liquid crystals with a single order parameter: i.e. *C*_26_(***r***) ∼ |***r***|^−*η*26^, with *η*_26_ = 6*k_B_T/*(*πK*), being *K* the orientational stiffness of both phases.

For finite hexatic and nematic activity, *C*_26_(***r***) exhibits instead an damped oscillatory behavior, marking the existence of a hierarchy of orientationally ordered structures, resulting from the hexatic and nematic activity, nested into each other at different length scales. In order to convey this new evidence we have majorly revised the section Multiscale order in epithelia, added a new figure (Figure 4), as well as a new appendix (Appendix 6), with all the analytical details.

8) Although emphasizing multiscale aspects, the model crucially neglects any mechanism of feedback. Active couplings other than the active stress, e.g., nonreciprocal cross terms aligning the two order parameters, are neglected, though they are likely to be more dominant in a gradient expansion compared to the hexatic contribution to the active stress.Furthermore, while substrate friction is included, traction forces (polar active forces) that are commonly exerted by cells are also neglected, which could presumably also dominate over hexatic degrees of freedom, even when randomized.

Feedback is in fact included via *five* different mechanisms, corresponding to *six* different terms in Equation (2). These are advection, precession and reorientation by the active flow, respectively embodied by the terms v⋅∇Qp,p[[Qp⋅ω]],λp[[∇⨂(p−2)u]] and vp[[∇⨂(pmod2)u⨂⌊p/2⌋]] in Eq. (2c), as well as the two equilibrium couplings κ2,6|Q2|2|Q6|2 and X2,6Q2⨂3⨀Q6 featured in the free energy *f*_2*,*6_ in Eq. (6b). As stressed in the Section The Model, while some of these terms drops in our analytical calculation of the structure factor, as a consequence of the linearization and of a few simplifying assumptions, they are fully accounted for in our numerical simulations. Higher order equilibrium couplings can, of course, be constructed by combining *Q*_2_ and *Q*_6_ with their derivatives. E.g. (Q2⨀∇Q2)⋅(Q6⨀∇Q6),|∇(Q2⨂3⨀Q6)|2,∇2(Q2⨂3⨀Q6) etc. Now, if the real and imaginary parts of the complex *p*−atic order parameter Ψp =⟨ψp⟩ are treated as independent variables, as we do in all our numerical calculations, these terms are also independent and must be separately included in the free energy *f*_2*,*6_. On the other hand, similarly to the zero-th order aligning interactions already included in our analysis – but less prominently, because of to the higher differential order – these couplings are expected to affect intermediate length scales, without interfering with neither the small scale hexatic structures, nor the large scale nematic behavior. For simplicity, we have decided to ignore these terms and focus on the already rich physics entailed in Equation (2).

Furthermore, like any theory built around an active stress, ours too is firmly rooted in Newton’s third law. As discussed in the text accompanying Equation (4) and (5), the construction of the active stress ***σ***^(a)^ is based on the assumption that all forces exerted by an active volume element on its surrounding are counterbalanced by equal and opposite forces, so that ∇⋅σ(a) =⟨∑cFc⟩, with ***F**_c_* the total force exerted by the *c*−th cell. In this respect, augmenting Eq. (2c) with non-reciprocal terms, originating from microscopic torques that violate Newton’s third law, would be manifestly inconsistent with the most basic assumptions of the theory itself.

Finally, whereas we have no doubt that the interplay between cells and the substrate are far more complex than anything that one can model via a simple frictional interaction, we are skeptical that adding this additional complication to the many already in place in this manuscript would provide a good service to the reader and to our understanding of collective behavior in epithelia. The goal of this work is to lay down the foundations of a theory that, we hope, will help us addressing several specific problems in the near future, not to solve all these problems simultaneously.

In response to comment #8, we have expanded the section *The Model* with several comments to clarify the scope and the limitations of the present model. Moreover, to place more emphasis on the role of the orientational coupling embodied in the parameter *χ*_2*,*6_, we now provide in Appendix 4 the full asymptotic expansion of the structure factor, inclusive of the coupling constants *χ_p_* = Γ*_p_χ*_2*,*6_, which were previously accounted for in the plots, but not in the analytical expression of the coefficient *s*_−2_ in Eq. (8).

9) Equation 3-5: It would be helpful if the notation could be fixed and either σ^α^ or \Pi is used. The relationship between the two is quite confusing at the moment.Also, sign conventions should be explained: is σ^α^ contractile, and if so then does the relation with f have an extra minus sign? Also, Eq. 5 suggests that α_6_ should have the same sign as α_2_, yet in the simulations opposite signs are chosen. What is its meaning and the relation between isotropic/hexatic and nematic extensile or contractile activity?It would also help to point out how the ratio of l2 and l6 (Eq. 7) differ by a ratio of the cell size to the conventional active nematic length (l2).The sign of ξ2,6 affects the alignment between Q2 and Q6, how does its sign affect the density and orientational correlations?

The use of different symbols for the active stress was meant to highlight the concept expressed in our reply to the previous point: all forces exerted by an active volume element on its surrounding (indicated with Π) are counterbalanced by equal and opposite forces (indicated with *σ*^(a)^). We now realize that this may be more confusing than clarifying and we have now eliminated the Π notation. This has further adjusted the issue with the sign of *α*_2_ and *α*_6_, which is now consistent with standard conventions.

The relation between nematic and hexatic activity is almost certainly non-trivial. In the simple analytical calculations anticipating Equation (5), we used the same symbols – i.e. *f* and *a* – to indicate the forces actively exerted at the scale of the cell, regardless on whether these are organized in dipoles (thus sourcing uniaxial active stresses) or hexapoles. Our experimental findings, however, suggest that nematic order originates from force chains propagating throughout the cellular layer in a way not dissimilar to granular materials, but with the essential differences that, while in granular materials these structures appear upon jamming as a reaction to the external confinement, in collectively migrating epithelia forces are internally generated, while the cellular layer is in a fluid state. For this reason, we believe that *α*_2_ is indeed related to *α*_6_, but this relation may be non-trivial and scale dependent, thus better suited to a Renormalization Group analysis. We are currently working on this problem and we expect to be able to offer a more detailed picture in the time scale of one year. For the same reason, we prefer to not venture is drawing specific conclusions about the ratio ℓ_2_/ ℓ_6_. We have commented on these caveats below Equation (5) and (7). Finally, as we now show in Appendix 6, the sign of *χ*_2*,*6_ determines where the hexatic and nematic orientations are preferentially parallel or antiparallel (i.e. tilted by 30^◦^), but have no prominent effects of the fluctuations.